# Inhibition of mitochondrial folate metabolism drives differentiation through mTORC1 mediated purine sensing

Martha M. Zarou [1], Kevin M. Rattigan [1], Daniele Sarnello [1], Engy Shokry[2], Amy Dawson[1], Angela Ianniciello[1], Karen Dunn [3], Mhairi Copland [3], David Sumpton [2], Alexei Vazquez [1] & G. Vignir Helgason [1]

Supporting cell proliferation through nucleotide biosynthesis is an essential requirement for cancer cells. Hence, inhibition of folate-mediated one carbon (1C) metabolism, which is required for nucleotide synthesis, has been successfully exploited in anti-cancer therapy. Here, we reveal that mitochondrial folate metabolism is upregulated in patient-derived leukaemic stem cells (LSCs). We demonstrate that inhibition of mitochondrial 1C metabolism through impairment of de novo purine synthesis has a cytostatic effect on chronic myeloid leukaemia (CML) cells. Consequently, changes in purine nucleotide levels lead to activation of AMPK signalling and suppression of mTORC1 activity. Notably, suppression of mitochondrial 1C metabolism increases expression of erythroid differentiation markers. Moreover, we find that increased differentiation occurs independently of AMPK signalling and can be reversed through reconstitution of purine levels and reactivation of mTORC1. Of clinical relevance, we identify that combination of 1C metabolism inhibition with imatinib, a frontline treatment for CML patients, decreases the number of therapy-resistant CML LSCs in a patient-derived xenograft model. Our results highlight a role for folate metabolism and purine sensing in stem cell fate decisions and leukaemogenesis.

Rapidly dividing cells require building blocks to sustain growth and proliferation. Folate metabolism, through a series of complementary cytosolic and mitochondrial reactions, provides 1C units for biosynthetic processes including nucleotide synthesis and the methionine cycle (Fig. 1a)[1,2]. New 1C units are donated from amino acids such as serine, the primary 1C unit donor, which gets converted into glycine via the enzyme serine hydroxymethyltransferase (SHMT)[1,3,4]. In parallel, activated 1C units are carried via folate molecules, allowing their assembly in different oxidative states[5].

In agreement with its key role for nucleotide synthesis, mitochondrial SHMT2 and the immediately downstream 5,10-methylene-tetrahydrofolate dehydrogenase (MTHFD2), are consistently overexpressed in cancer[6,7]. Furthermore, serine catabolism to formate is increased in intestinal adenomas and mammary carcinomas[8]. Reinforcing of the requirement for folate metabolism in cancer, antifolates have proven highly effective as chemotherapeutic agents. Methotrexate, the most recognised antifolate, has been used in the treatment of a variety of neoplasms and inflammatory diseases[9,10]. Importantly, antifolates have found a wide application in haematological malignancies[11,12]. However, the role of folate metabolism in LSCs remains undescribed.

---

[1]Wolfson Wohl Cancer Research Centre, School of Cancer Sciences, University of Glasgow, Glasgow G61 1QH, UK. [2]Cancer Research UK Scotland Institute, Glasgow G61 1BD, UK. [3]Paul O'Gorman Leukaemia Research Centre, School of Cancer Sciences, University of Glasgow, Glasgow G12 0ZD, UK. ✉e-mail: avazque1@protonmail.com; vignir.helgason@glasgow.ac.uk

CML is a stem cell-driven myeloproliferative disorder that arises in a single haematopoietic stem cell (HSC) with the introduction of the BCR::ABL1 oncoprotein[13]. BCR::ABL1, through activation of several molecular pathways including RAS, MYC and PI3K/AKT/mTOR, is the main driver of leukaemia initiation and progression[14–16]. As exemplified in CML, increasing evidence suggests that stem cells, but not fully differentiated cells, are a major driver of neoplasms and therapy resistance[17–19]. Oncogenic stem cells evade standard therapy due to their quiescent/slow-cycling status[20]. To maintain tissue homeostasis, stem cells, including HSCs, are characterised by their capacity to self-renew and their ability to differentiate[21,22]. Thus, identification of vulnerabilities that when targeted, can stimulate differentiation of stem/progenitor cell population can have potent anticancer effects[23–25].

Specifically, slow-growing CML LSCs have been found to resist treatment with tyrosine kinase inhibitors (TKIs), even when BCR::ABL1 kinase activity is fully inhibited[26–28]. It has been established that BCR::ABL1 rewires metabolic programmes such as oxidative phosphorylation (OXPHOS) and the tricarboxylic acid cycle[29,30]. As a result, targeting OXPHOS in combination with standard therapy was found to eradicate persistent LSCs[29]. Furthermore, it was recently demonstrated that exogenous serine controls the cell fate of epidermal stem cells during tumour initiation[31]. These studies highlight how metabolic rewiring of oncogenic stem cells might affect tumour initiation and persistence.

In the present study, we describe deregulation of mitochondrial folate metabolism in leukaemic stem and progenitor cells. We demonstrate that inhibition of mitochondrial folate metabolism through impairment of de novo purine synthesis drives differentiation of CML cells. Mechanistically, we show that the differentiation following nucleotide depletion depends on suppression of mTORC1 activity through decreased purine sensing. Moreover, we uncover that inhibition of 1C metabolism sensitises human LSCs to standard CML therapy. Collectively, our results provide insight into the role of folate metabolism in critical cell fate decisions.

## Results

### Mitochondrial folate metabolism is deregulated in LSCs and necessary for tumour formation in vivo

To assess whether folate metabolism is deregulated in primitive CML cells, we carried out differential gene expression analysis of a transcriptomic dataset generated from CML LSCs (CD34$^+$CD38$^-$) and normal HSCs (CD34$^+$CD38$^-$)[32]. Gene Set Enrichment Analysis (GSEA) revealed that folate metabolism is upregulated in LSCs (Supplementary Fig. 1a, b). Specifically, there was an upregulation of gene transcripts encoding enzymes involved in mitochondrial 1C reactions such as *ALDH1L2*, *SHMT2* and *MTHFD2* (Fig. 1b and Supplementary Fig. 1c). The upregulation of mitochondria-related genes supports the observation that most cancer cells produce 1C units exclusively in the mitochondria[5]. To confirm that folate metabolism is deregulated in primitive CML cells, we functionally determined the activity of folate metabolism in patient-derived CML CD34$^+$ cells and normal counterparts. To this end, using gas chromatography-mass spectrometry (GS-MS) we assessed formate overflow as a proxy for the rate of serine catabolism. CML CD34$^+$ cells had an increased formate exchange rate and extracellular formate concentration compared to normal CD34$^+$ cells, confirming a higher rate of folate metabolism in CML cells (Fig. 1c and Supplementary Fig. 1d).

We next analysed the consequences of 1C metabolism inhibition in three CML cell lines, K562, KCL22 and JURL-MK1. We used the pharmacological SHMT1/2 inhibitor SHIN1 (which inhibits cytosolic and mitochondrial arms of the pathway)[33], in parallel with a genetic knockout (KO) of mitochondrial SHMT2 (Supplementary Fig. 2b, f, i). Due to the reversibility of the cytosolic compartment of folate metabolism, mitochondrial 1C metabolism is believed to be dispensable[5,34]. As such, single deletions of mitochondrial enzymes do not consistently

block cancer cell growth in nutrient rich conditions[5,34,35]. Notably, we observed a proliferation halt following either pharmacological (cytosolic/mitochondrial) or genetic (mitochondrial) inhibition of 1C metabolism (Fig. 1d, e and Supplementary Fig. 2a–j). Furthermore, SHIN1 treatment or SHMT2 KO resulted in accumulation of K562 cells in the S phase of the cell cycle (Fig. 1f and Supplementary Fig. 2k, l). Formate supplementation was sufficient to restore growth, indicating that 1C units are the limiting factor for proliferation and cell cycle progression of CML cells (Fig. 1d–f and Supplementary Fig. 2a–l). To assess whether purine or thymidylate depletion caused the proliferation arrest following inhibition of 1C metabolism, we supplemented cells with the purine salvage precursor hypoxanthine, thymidine or a combination of both. While hypoxanthine rescued proliferation, thymidine did not (Fig. 1d–f and Supplementary Fig. 2a–l). These data are consistent with the fact that purine biosynthesis requires a higher flux of 1C units compared to thymidylate synthesis[36]. It is likely that in the short-term, residual 1C units can sustain thymidylate synthesis. Our findings support that nucleotide impairment following pharmacological or genetic inhibition of folate metabolism has a cytostatic effect on CML cells.

Following the observation that mitochondrial 1C metabolism inhibition results in proliferation arrest in CML cells, we sought to evaluate the reversibility of the cytosolic 1C pathway in CML SHMT2 KO cells. To this end, we assessed the incorporation of deuterium labelled 2,3,3-$^2$H$_3$-serine into deoxythymidine trisphosphate (dTTP). Deuterium labelling allows for the compartment-specific analysis of the flux of 1C units. Transfer of 1C units from the mitochondrion to the cytosol results in thymidylate with a single deuterium atom (m + 1), whereas the cytosolic route produces m + 2 thymidylate (Fig. 1g). SHMT2 competent control CML cells predominantly displayed m + 1 dTTP, suggesting that most of 1C units originate in the mitochondria, whereas loss of SHMT2 resulted in majority m + 2 dTTP and almost complete depletion of the m + 1 dTTP isotopologue (Fig. 1h and Supplementary Fig. 2m). Similar adaptation to loss of mitochondrial 1C units was observed in the colorectal HCT116 SHMT2 KO cells (Supplementary Fig. 2n, o). Having established that CML cells can switch to cytosolic dTTP production upon loss of mitochondrial serine catabolism, we assessed whether this stands true for newly synthesised purine nucleotides. To explore this, we used stable isotope labelled $^{13}$C$_3$$^{15}$N$_1$-serine and measured its incorporation to ATP. Interestingly, while genetic ablation of SHMT2 did not change the ability of HCT116 cells to synthesise ATP, it dramatically decreased the incorporation of serine into ATP in all three CML cell lines, indicating that mitochondrial folate metabolism is indispensable for purine synthesis in CML cells (Fig. 1i and Supplementary Fig. 2p).

CML KCL22 cells form extramedullary tumours when transplanted in immunocompromised mice. To test whether loss of the mitochondrial folate cycle impairs tumour formation in vivo, KCL22 SHMT2 KO cells were transplanted via tail vain injection into non-irradiated NRGW (NOD.Cg-*Rag1*$^{tm1Mom}$*Kit*$^{W-4Jl}$*Il2rg*$^{tm1Wjl}$)[41] mice. Loss of SHMT2 significantly prolonged tumour-free survival of xenografted mice, when compared with mice receiving SHMT2 competent control cells (Fig. 1j). This correlated with a significant decrease in tumour burden in mice transplanted with KCL22 SHMT2 KO cells (Fig. 1k). Together, these results suggest that mitochondrial 1C metabolism sustains proliferation of CML cells both in vitro and in vivo, and cytosolic 1C metabolism does not fully compensate for the loss of SHMT2.

### Inhibition of mitochondrial serine catabolism disrupts de novo purine synthesis and glycolysis in CML cells

Next, we sought to investigate further the downstream metabolic effects of folate cycle suppression. As expected, metabolic analysis revealed that SHIN1 treatment or SHMT2 deficiency led to reduction of the glycine pool (Fig. 2a and Supplementary Fig. 3a). Moreover,

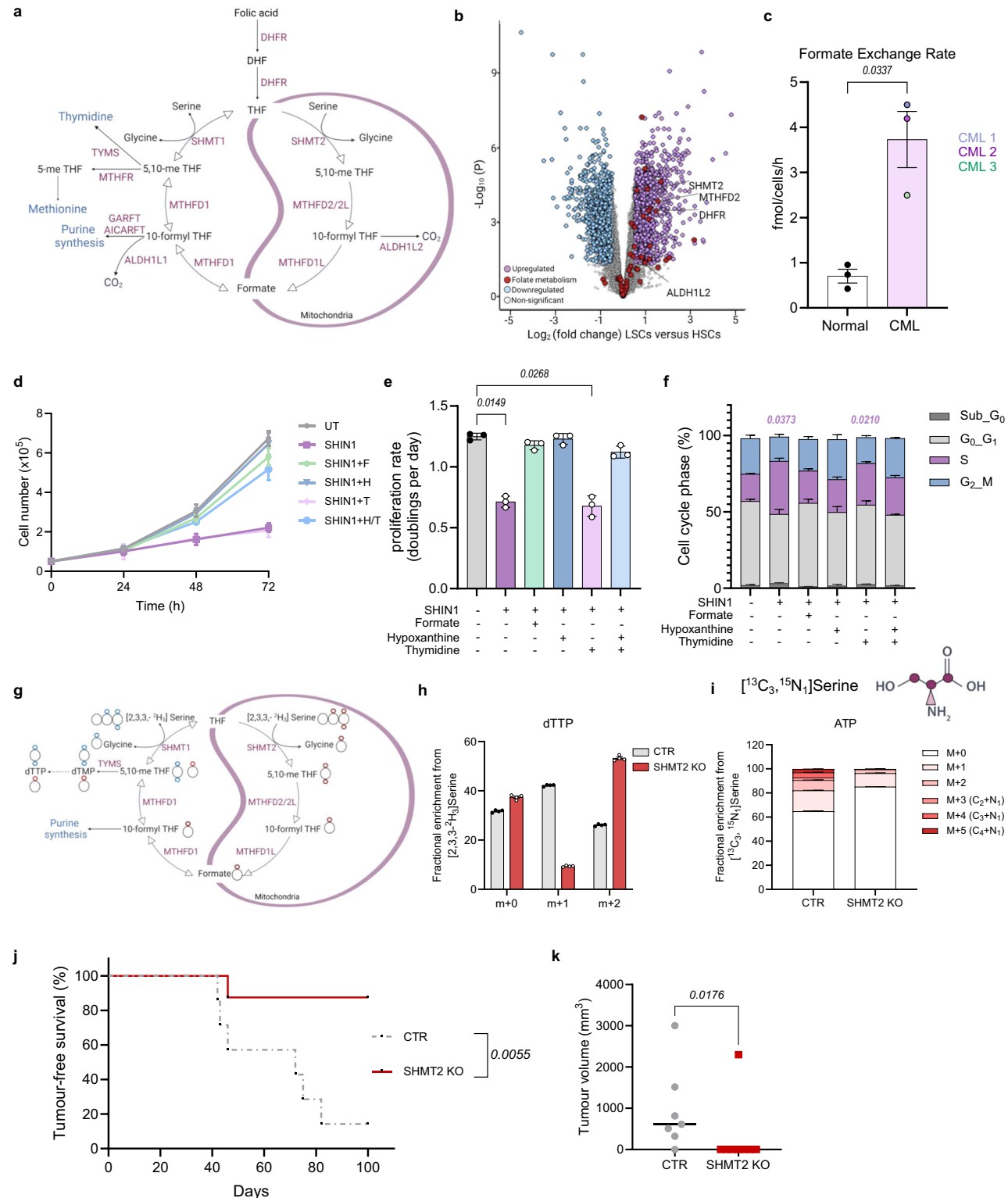

formate supplementation resulted in a further decrease of glycine levels, as it restores the 1C unit pool and drives de novo purine synthesis bypassing SHMT1/2 activity. Notably, we observed a dramatic increase in the purine biosynthetic intermediate 5'-phosphoribosyl-5-aminoimidazole-4-carboxamide (AICAR), and a corresponding reduction in purine nucleotides (Fig. 2c and Supplementary Fig. 3c). Changes in purine nucleotide pools were reversed by formate. Surprisingly, we observed a reduction in the ribose-5-phosphate (R5P) pool, which is upstream of a step requiring glycine.

R5P, apart from being an intermediate of purine/pyrimidine synthesis, is a product of the non-oxidative phase of the pentose phosphate pathway (Fig. 2b and Supplementary Fig. 3b). Prompted by this reduction, we decided to investigate the effect of 1C metabolism inhibition on glycolysis related metabolites. Pharmacological or genetic inhibition of 1C metabolism resulted in increased intracellular glucose levels and decreased pyruvate and lactate levels (Fig. 2d and Supplementary Fig. 3d). Subsequently, quantitative analysis of the media from SHIN1-treated or SHMT2 KO cells confirmed a significant

**Fig. 1 | Folate metabolism is deregulated in LSCs and necessary for tumour formation in vivo. a** Schematic showing the compartmentalised folate metabolism pathway. Created with Biorender.com (Agreement number IY26CFF7VX). **b** Volcano plot of differentially expressed genes in CML LSCs compared to normal HSCs (CD34$^+$CD38$^-$) (E-MTAB-2581). Upregulated genes with a log$_2$ (fold change) <0.5 are shown in light purple, downregulated genes with a log$_2$ (fold change) < −0.5 are in blue. Folate metabolism associated genes are in red. **c** Formate release rate from normal and CML CD34$^+$ cells following 48 h culture ($n$ = 3 patient samples). Growth (**d**) and proliferation rate (**e**) of K562 untreated (UT) or cultured with 2.5 μM SHIN1 with or without 1 mM formate (F), 100 μM hypoxanthine (H) and 16 μM thymidine (T) for 72 h ($n$ = 3 independent cultures). **f** Percentage of cell cycle phases of K562 cells treated as in (**d**) ($n$ = 3 independent cultures). **g** Schematic of 2,3,3-$^2$H$_3$-serine incorporation into deoxythymidine triphosphate (dTTP). Coloured dots represent cytosolic or mitochondrial derived $^2$H atoms in individual metabolites. Created with Biorender.com (Agreement number RW26CFFSYQ). **h** Relative mass isotopologue distribution in dTTP from K562 control (CTR) and SHMT2 knockout (KO) cells cultured with 140 μM 2,3,3-$^2$H$_3$-serine ($n$ = 4 independent wells from individual experiment). **i** Fractional enrichment of ATP from K562 CTR and SHMT2 KO cells cultured with 140 μM $^{13}$C$_3$$^{15}$N$_1$-serine. ($n$ = 4 independent wells from individual experiment). Schematic of serine created with Biorender.com (Agreement number EZ26CFMM6G). Tumour-free survival (**j**) and tumour volume (**k**) at endpoint of NRGW[41] mice transplanted with KCL22 CTR ($n$ = 7 mice) and SHMT2 KO cells ($n$ = 8 mice). Data are shown as the mean ± s.e.m. $P$-values were calculated using DESeq2 two-tailed $t$-test with Benjamini–Hochber's multiple comparisons test (**b**), unpaired two-tailed $t$-test with Welch's correction (**c**), a repeated measure one-way ANOVA with Dunnett's multiple comparisons test (**e**), two-way ANOVA with Tukey's multiple comparisons test (**f**), log-rank (Mantel–Cox) test for survival analysis (**j**) or unpaired two-tailed Mann–Whitney test (**k**). $P$-values in magenta (**f**) are derived from comparing the percentage of cells in the S phase between UT, SHIN1 and SHIN1 + T. Source data are provided as a Source Data file.

reduction in glucose uptake and lactate secretion, when compared with media from 1C metabolism competent cells (Fig. 2e and Supplementary Fig. 3e). To confirm the decrease in glycolytic flux, we measured the extracellular acidification rate (ECAR) following suppression of 1C metabolism. SHIN1 treatment or SHMT2 ablation reduced both glycolysis and glycolytic capacity (Fig. 2f, g and Supplementary Fig. 3f, g). Of note, formate supplementation reversed the decrease in glycolytic flux, suggesting that impaired glycolysis was linked to reduced nucleotide levels. This is in agreement with our previous work where we applied theoretical modelling to indicate that 1C metabolism promotes glycolysis[37].

### Suppression of mitochondrial folate metabolism leads to differentiation of CML cells in an AMPK-independent manner

Since AICAR and purine nucleotides can directly or indirectly signal to AMP-activated protein kinase (AMPK)[38], we next sought to investigate the effect of mitochondrial folate metabolism inhibition on AMPK activity. Indeed, SHIN1 treatment and SHMT2 KO induced AMPK activity, measured by increased phosphorylation of AMPK on threonine-172, and increased phosphorylation of the two AMPK effectors, acetyl-coA carboxylase (ACC) and Unc-51 like autophagy activating kinase 1 (ULK1) in K562 cells (Fig. 3a and Supplementary Fig. 4a). Increased ULK1-dependent phosphorylation of the autophagy related 13 (ATG13) protein suggests activation of the autophagy initiation complex (Fig. 3a and Supplementary Fig. 4a). Concurrently, pharmacological or genetic inhibition of folate metabolism reduced phosphorylation of the ribosomal protein S6 (RPS6), a target of the mechanistic target of rapamycin complex 1 (mTORC1) (Fig. 3a and Supplementary Fig. 4a). Treatment with AICAR resulted in a similar pattern of AMPK signalling activation and mTORC1 suppression, while combination of AICAR with folate metabolism inhibition did not have any additive effect (Fig. 3a and Supplementary Fig. 4a). Additionally, formate supplementation reversed the effect on AMPK and mTORC1 (Fig. 3a and Supplementary Fig. 4a). Similar pattern of increased AMPK phosphorylation and reduced phosphorylation of RPS6 was observed in CML KCL22 and JURL-MK1 cells following SHIN1 treatment (Supplementary Fig. 4b, c).

In addition to being an activator of AMPK, AICAR has been shown to promote differentiation of acute myeloid leukaemia (AML) cells[39,40]. Thus, we next assessed whether inhibition of mitochondrial folate metabolism induces differentiation of CML K562 cells. These cells have been shown to be a suitable model to study differentiation, as they undergo erythroid maturation when treated with differentiation-inducing compounds[41]. Macroscopic examination revealed that cell pellets appeared with a strong red colour following inhibition of mitochondrial serine catabolism, which suggests a higher haemoglobin production and, ultimately, an increase in erythroid maturation (Supplementary Fig. 4d). To confirm initiation of erythroid maturation, we stained K562 cells for erythropoiesis markers: Glycophorin A

(GlyA), a sialoglycoprotein found in the membrane of erythrocytes, and transferrin receptor 1 (CD71), a transmembrane glycoprotein necessary for iron import inside the cells[42]. SHIN1 treatment resulted in enhanced levels of GlyA and CD71 (Fig. 3b, c). Similarly, cell surface expression of GlyA and CD71 increased following loss of SHMT2 (Supplementary Fig. 4e). Of note, AICAR treatment promoted expression of GlyA and CD71 to the same extent as SHIN1, while we did not observe any additive effect when combining AICAR with SHIN1 (Fig. 3b, c). Restoration of the 1C unit pool through formate supplementation was sufficient to reverse expression of erythroid markers (Fig. 3b, c). It has been reported that differentiation stimuli can increase surface expression of GlyA in JURL-MK1 cells[43]. Consistently, both SHIN1 and AICAR treatment increases expression of CD71 and GlyA in JURL-MK1. The addition of formate reversed enhanced levels of maturation markers, as seen on K562 cells (Supplementary Fig. 4f). To further assess the role of folate metabolism in differentiation of CML cells, we cultured patient-derived CML CD34$^+$ cells with low concentration (3U/ml) of erythropoietin, required to sensitise cells to erythroid commitment. As observed with K562 and JURL-MK1 cells, SHIN1 treatment increased levels of GlyA and CD71 in four separate CML CD34$^+$ patient samples, while formate supplementation reversed enhanced expression of erythroid markers (Fig. 3d). As AICAR was found to induce differentiation of AML cells[39,40], we hypothesised that folate metabolism inhibition would elicit a similar effect on AML cells. To address this, we took advantage of two monocytic AML cell lines, THP1 and MOLM13, which express monocytic surface markers such as CD11b. Increased expression of CD11b correlates with differentiation into the myeloid lineage. Pharmacological inhibition of 1C metabolism resulted in increased expression of CD11b, indicating that SHIN1 treatment induces myeloid differentiation in AML cells, and, therefore, the effect is not specific to K562 and JURL-MK1 cells and erythroid differentiation (Supplementary Fig. 4g).

Maturation of progenitor cells has been associated with increased oxidative stress and reactive oxygen species (ROS)[44,45]. Therefore, we measured levels of mitochondrial superoxide following SHIN1 treatment. Surprisingly, SHMT1/2 inhibition resulted in decreased mitochondrial ROS levels (Fig. 3e). Moreover, addition of the ROS scavenger N-acetylcysteine (NAC) did not reverse the enhanced expression of GlyA and CD71 in SHIN1 treated cells (Fig. 3f). These data suggest that initiation of differentiation following inhibition of folate metabolism is not a result of increased ROS production. We next sought to investigate whether AMPK activation was playing a role in differentiation of CML cells. We generated AMPKα1 AMPKα2 double-knockout (DKO) cells (Supplementary Fig. 4h), and measured levels of erythroid markers following incubation with either SHIN1 or AICAR. Surprisingly, there was no significant difference in expression of erythroid markers between SHIN1-treated control and AMPKα DKO cells, indicating that initiation of differentiation is not dependent on AMPK activity (Fig. 3g). Intriguingly, inhibition of folate metabolism suppressed mTORC1 activity in

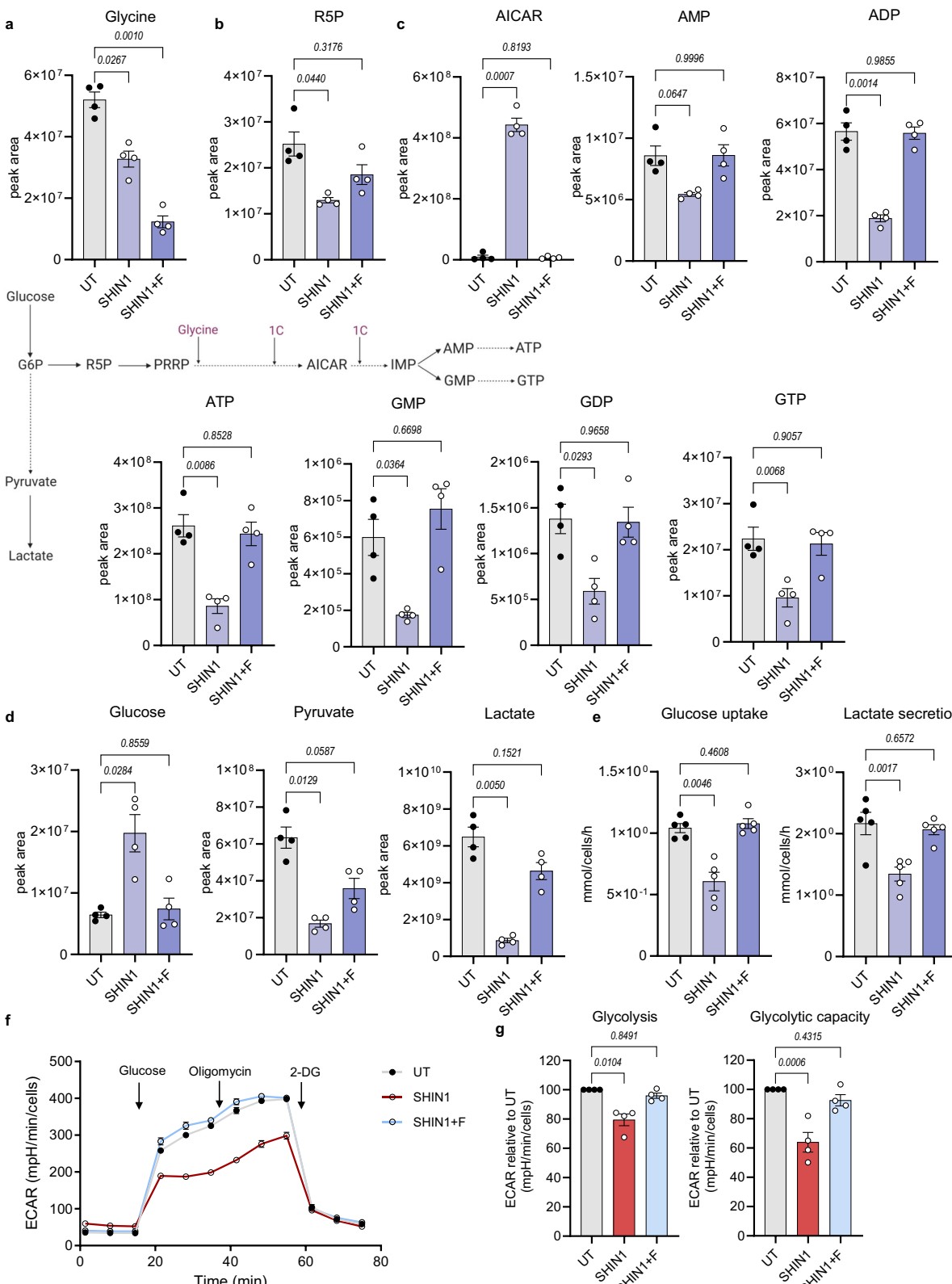

**Fig. 2 | Inhibition of mitochondrial serine catabolism disrupts de novo purine synthesis and glycolysis in CML cells.** LC-MS measurement of intracellular glycine (**a**), ribose-5-phosphate (R5P) (**b**), purine biosynthetic intermediate and nucleotides (**c**) and glycolysis related metabolites (**d**) in K562 cells treated with 2.5 μM SHIN1 with or without the addition of 1 mM formate for 24 h (*n* = 4 independent cultures). **e** Glucose uptake and lactate secretion in K562 cells in response to treatment as explained in (**a**–**d**) (*n* = 5 independent cultures). Representative extracellular

acidification rate (ECAR) profile (**f**), relative glycolysis and glycolytic capacity (**g**) as measured by the Seahorse XF analyser of K562 cells treated as in (**a**–**e**) (*n* = 4 independent cultures). Data are shown as the mean ± s.e.m. *P*-values are derived from a repeated measure one-way ANOVA with Dunnett's multiple comparisons test (**a**–**e**) or ordinary one-way ANOVA with Dunnett's multiple comparisons test (**g**). Source data are provided as a Source Data file.

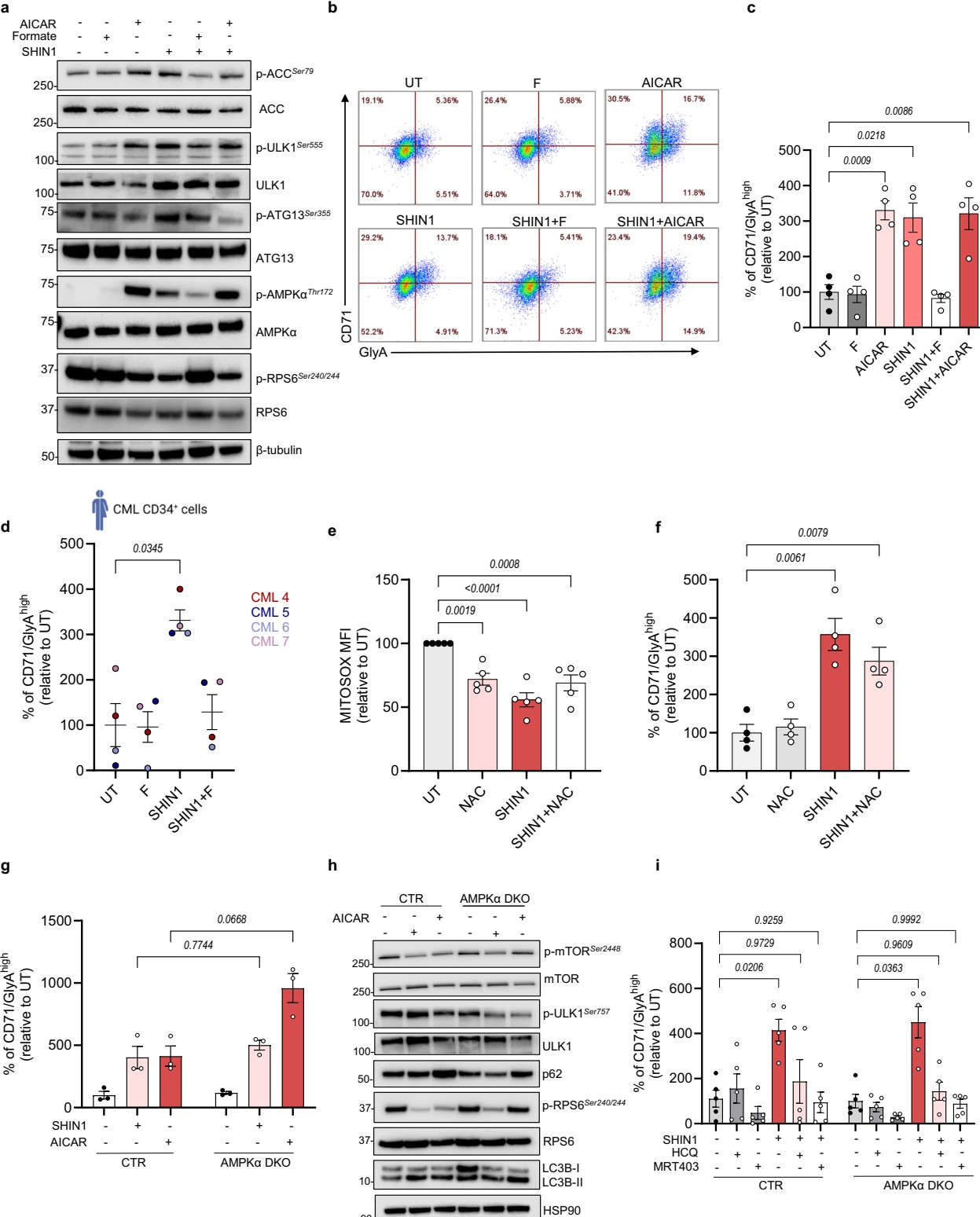

AMPKα DKO cells, measured by decreased phosphorylation of ULK1 on serine-757 and RPS6 (Fig. 3h). The effect of AICAR on mTORC1 activity was reversed in AMPKα DKO cells, suggesting that AMPK signalling is required for AICAR-induced mTORC1 suppression (Fig. 3h).

mTORC1 and AMPK integrate energy signals to balance anabolic and catabolic pathways, including autophagy. When assessing levels of autophagy markers in AMPKα DKO cells, we observed that SHIN1 promoted reduction in the autophagy cargo receptor sequestosome 1 (SQSTM1/p62) and accumulation of the microtubule-associated protein 1 light chain 3B (LC3B-II), indicating induction of the autophagic flux (Fig. 3h). To assess whether autophagy inhibition could reverse the SHIN1-driven differentiation, we utilised two autophagy inhibitors, hydroxychloroquine (HCQ), which blocks the fusion of the autophagosome with the lysosome[46], and a recently developed inhibitor of ULK1, MRT403[25]. HCQ and MRT403 were sufficient to reverse SHIN1-driven differentiation in control and AMPKα DKO cells (Fig. 3i).

**Fig. 3 | Suppression of mitochondrial folate metabolism leads to ROS and AMPK-independent differentiation of CML cells. a** Immunoblot analysis to assess AMPK and mTORC1 signalling in K562 cells following 24 h incubation with 1 mM AICAR, 2.5 μM SHIN1 in the presence or absence of 1 mM formate and combination of SHIN1 with AICAR (representative of three independent experiments). **b** Representative flow cytometry plots of CD71 and GlyA expression in K562 cells subjected to treatment as in (**a**). **c** Percentage of CD71/GlyA^high cells relative to untreated as depicted in (**b**) (n = 4 independent cultures). **d** Percentage of CD71/ GlyA^high CML CD34^+ cells relative to untreated subjected to treatment as in (**a**). Cells were incubated in the presence of 3 U/ml erythropoietin (n = 4 patient samples). Schematic for patient cells created with Biorender.com (Agreement number SH26CFO8OE). **e** Quantification of relative mean fluorescence intensity (MFI) of mitochondrial ROS in K562 cells treated with 2.5 μM SHIN1 with or without 300 μM NAC for 72 h (n = 5 independent cultures). **f** Percentage of CD71/GlyA^high K562 cells relative to untreated after 72 h incubation with treatments as in (**e**) (n = 4

independent cultures). **g** Relative percentage of CD71/GlyA^high K562 CTR and AMPKα DKO cells in the presence of 2.5 μM SHIN1 or 1 mM AICAR for 72 h (n = 4 independent cultures). **h** Immunoblot analysis to assess mTORC1 activity and autophagy induction in K562 CTR and AMPKα DKO cells following 24 h culture with 1 mM AICAR or 2.5 μM SHIN1 (representative of three independent experiments). **i** Relative percentage of CD71/GlyA^high K562 CTR and AMPKα DKO cells treated with 2.5 μM SHIN1 with or without 5 μM hydroxychloroquine (HCQ) or 3 μM MRT403 for 72 h (n = 5 independent cultures). Data are shown as the mean ± s.e.m. P-values were calculated with a repeated measure one-way ANOVA with Dunnett's multiple comparisons test (**c**, **f**, **i**), Kruskal–Wallis test with Dunn's multiple comparisons test (**d**), ordinary one-way ANOVA with Dunnett's multiple comparisons test (**e**) or two-way ANOVA with Sidak's multiple comparisons test (**g**). Exact p-value in (**e**) comparing UT with SHIN1 treated cells is 0.000017. Source data are provided as a Source Data file.

Furthermore, SHIN1 treatment did not significantly induce expression of GlyA and CD71 in autophagy deficient, ULK1 KO and autophagy related 7 (ATG7) KO cells (Supplementary Fig. 4i, j). Collectively, these data suggest that inhibition of mitochondrial 1C metabolism induces differentiation of CML cells in an AMPK independent manner, while autophagy inhibition reverses such effect.

## Impairment of purine biosynthesis drives differentiation of CML cells through inhibition of mTORC1 signalling

Having established that suppression of mitochondrial folate metabolism drives differentiation and suppression of mTORC1 activity independently of AMPK, we asked whether there is a link between mTORC1 signalling and differentiation. It has been previously described that mTORC1 senses acute changes in purine nucleotide levels, and more specifically adenylates[47,48]. To determine if this is the case in CML cells, we supplemented hypoxanthine in K562 SHIN1-treated or SHMT2 KO cells and assessed mTORC1 activity. As expected, SHIN1 treatment decreased phosphorylation of the direct mTORC1 kinase target S6 (S6K) and RPS6, while hypoxanthine rescued mTORC1 activity (Fig. 4a and Supplementary Fig. 5a). Importantly, short-term (1 h) supplementation of hypoxanthine was sufficient to reverse mTORC1 signalling, supporting the notion that mTORC1 can acutely sense changes in purine levels. As anticipated, mTORC1 was not reactivated following thymidine supplementation (Fig. 4a and Supplementary Fig. 5a). Furthermore, long-term (24 h) hypoxanthine supplementation repressed AMPK signalling in SHIN1-treated cells, whereas short-term hypoxanthine had a modest effect on AMPK activity (Supplementary Fig. 5b). The pattern of mTORC1 signalling strongly correlated with expression of erythropoiesis and myelopoiesis markers, with hypoxanthine reversing enhanced levels of GlyA and CD71 in CML cells (K562 and JURL-MK1) and CD11b in AML cells (THP1 and MOLM13) (Fig. 4b and Supplementary Fig. 5c, d). In addition to hypoxanthine, adenine treatment restored mTORC1 activity in SHIN1-treated or SHMT2 KO K562 cells (Fig. 4c and Supplementary Fig. 5e). Long-term guanine supplementation partially reactivated mTORC1, whereas short-term supplementation did not have any effect on mTORC1 signalling. Moreover, only adenine, but not guanine supplementation, repressed increased surface levels of erythroid or myeloid maturation markers following 1C metabolism inhibition (Fig. 4d and Supplementary Fig. 5f, g). Similarly, while long-term adenine supplementation fully restored mTORC1 signalling and reversed differentiation in AMPKα DKO, guanine partially reactivated mTORC1, which was not sufficient to reverse differentiation (Fig. 4e, f). To further assess whether mTORC1 inhibition was sufficient to induce differentiation, we treated CML cells with rapamycin and measured levels of erythropoiesis markers. Rapamycin enhanced levels of GlyA and CD71 to the same extent as SHIN1 (Supplementary Fig. 5h). Of note, combination of SHIN1 with rapamycin modestly increased levels of GlyA and CD71 compared to SHIN1 monotherapy, however this did not reach

statistical significance (Supplementary Fig. 5h). Ultimately, to confirm the role of mTORC1 activity in CML differentiation, we generated tuberous sclerosis complex 2 (TSC2) KO cells (Fig. 4g and Supplementary Fig. 5i). TSC2 is part of the tumour suppressor TSC complex that maintains mTORC1 inactive through the regulation of Ras homologue enriched in brain (Rheb). Additionally, TSC2 has been shown to be required for purine sensing by mTORC1[47]. Accordingly, loss of TSC2 diminished the effect of SHIN1 on mTORC1 signalling and differentiation in CML cells (Fig. 4g, h and Supplementary Fig. 5i, j). As expected, rapamycin suppressed mTORC1 activity and induced expression of erythroid maturation markers in TSC2 KO cells in line with Supplementary Fig. 5h. Conversely, SHIN1 suppressed mTORC1 activity and increased cell surface expression of CD11b in THP1 and MOLM13 TSC2 KO cells (Supplementary Fig. 5k–n). Surprisingly, while exposure to rapamycin inhibited mTORC1 signalling, it did not alter levels of CD11b in AML control or TSC2 KO cells (Supplementary Fig. 5k–n). Together, these data uncover that mitochondrial 1C metabolism inhibition uniquely mediates differentiation through modulation of the mTORC1 pathway in CML cells.

Having determined a link between inhibition of folate metabolism, purine sensing through mTORC1 and differentiation, we sought to further explore the mechanism behind AICAR-induced differentiation. Metabolomic analysis revealed that AICAR treatment significantly decreased the pyrimidine nucleotide pool, while it did not have any significant effect on purine nucleotides (Supplementary Fig. 6a, b). It is also worth noting that SHIN1 did not alter levels of pyrimidine nucleotides (Supplementary Fig. 6a). These findings are in agreement with previous literature suggesting that AICAR induces differentiation through pyrimidine depletion[40]. To further examine whether impairment of de novo pyrimidine synthesis results in differentiation of CML cells, we treated K562 cells with the uridine analogue 6-azauridine. 6-azauridine blocks the conversion of orotate to uridine monophosphate (UMP) (Fig. 4i). Importantly, impairment of pyrimidine biosynthesis enhanced expression of erythropoiesis markers (Fig. 4j). Uridine supplementation reversed differentiation, supporting that 6-azauridine treatment induced maturation through depletion of pyrimidine nucleotides (Fig. 4j). Moreover, 6-azauridine had a minimal effect on mTORC1 activity, measured by decreased phosphorylation of RPS6, although not to the same extent as SHIN1 (Fig. 4k). Pyrimidine depletion through 6-azauridine treatment mediated erythroid maturation in TSC2-deficient cells (Fig. 4l), suggesting that differentiation following pyrimidine impairment is not dependent on mTORC1 modulation. Overall, these results reveal that impairment of de novo nucleotide synthesis results in differentiation of CML cells.

## SHIN1 selectively targets LSCs in combination with standard CML therapy

As previously highlighted, CML is a paradigm for stem cell driven malignancies, as it originates in a single HSC following the generation

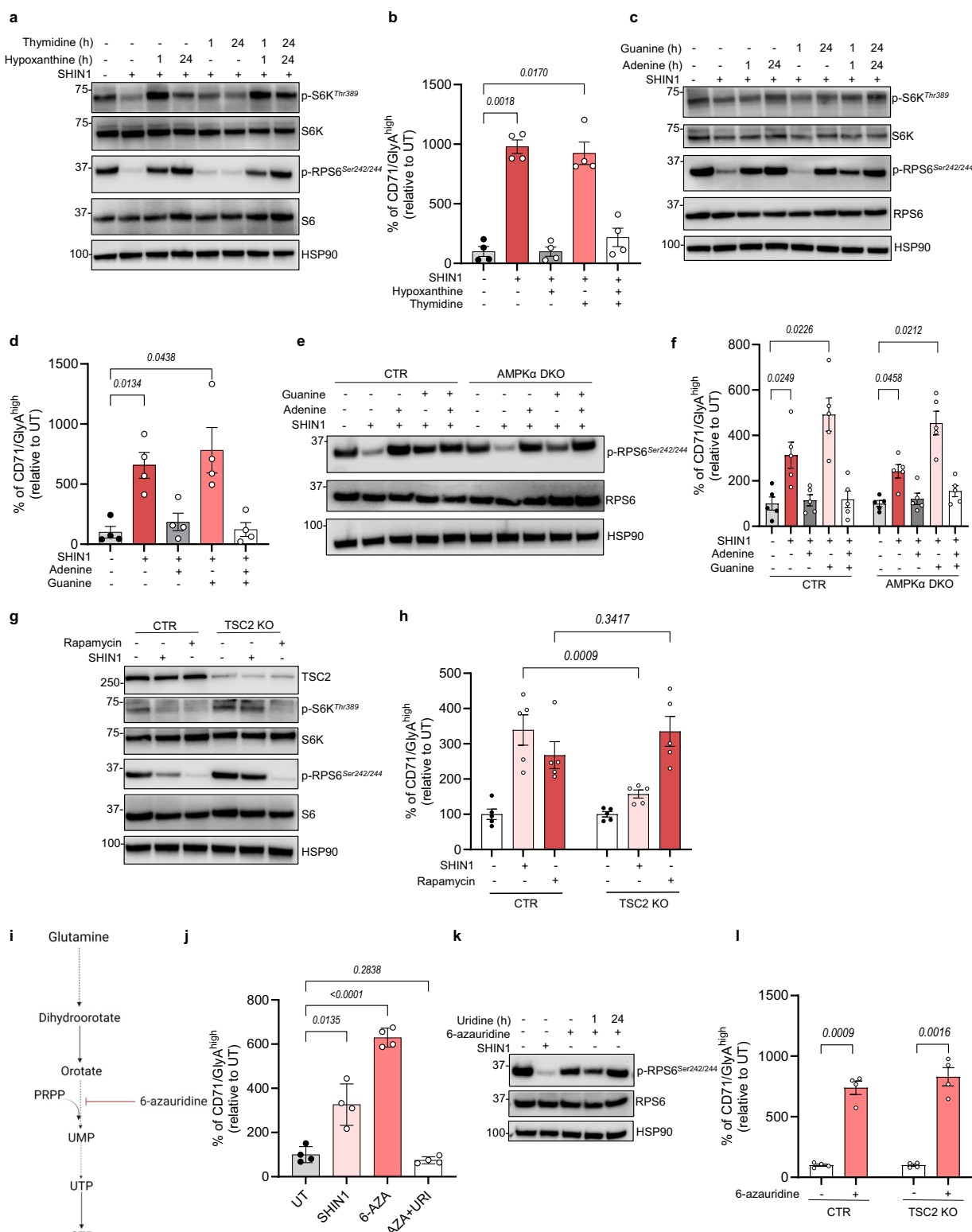

of the BCR::ABL1 fusion oncoprotein. While the introduction of TKIs such as imatinib has significantly improved response rates of CML patients, disease initiating LSCs are insensitive to TKI therapy[28]. Therefore, having established that folate metabolism is deregulated in LSCs (Fig. 1b), we sought to investigate the combinatory effect of SHIN1 and imatinib treatment on 1C metabolism and survival of primitive CML cells. First, we used stable isotope labelled $^{13}C_3^{15}N_1$-serine

to determine the effect of imatinib on serine metabolism in K562 cells. Due to SHMT reversibility, serine's labelling pattern consists of a mixture of $m+1$, $m+3$, $m+4$ isotopologues, reflecting the different recombination events with labelled and unlabelled glycine and 1C units derived from methylene-THF. As expected, SHIN1 treatment increased the abundance of $m+4$ serine isotopologue, while it dramatically decreased $m+1$ and $m+3$ fractions (Fig. 5a and Supplementary

**Fig. 4 | Impairment of nucleotide biosynthesis drives differentiation of CML cells. a** Immunoblot analysis of mTORC1 targets in K562 cells exposed to 2.5 µM SHIN1 with or without 100 µM hypoxanthine and/or 16 µM thymidine for the indicated times (representative of three independent experiments). **b** Relative percentage of CD71/GlyA$^{high}$ K562 cells exposed to treatments as in (**a**) for 72 h (n = 4 independent cultures). **c** Immunoblot analysis of mTORC1 targets in K562 cells treated with 2.5 µM SHIN1 with or without 30 µM adenine and/or 30 µM guanine for the indicated times (representative of three independent experiments). **d** Relative percentage of CD71/GlyA$^{high}$ K562 cells treated as depicted in (**c**) for 72 h (n = 4 independent cultures). **e** Immunoblot analysis of RPS6 phosphorylation in K562 CTR and AMPKα KO cells incubated with treatments as depicted in (**c**) for 24 h (representative of three independent experiments). **f** Relative percentage of CD71/GlyA$^{high}$ K562 CTR and AMPKα DKO cells treated as in (**d**) for 72 h (n = 5 independent cultures). **g** Immunoblot analysis of mTORC1 targets in K562 CTR and TSC2 KO cells incubated with 2.5 µM SHIN1 or 10 nM rapamycin for 24 h (representative of three

independent experiments). **h** Relative percentage of CD71/GlyA$^{high}$ K562 CTR and TSC2 KO cells following treatment as in (**g**) for 72 h (n = 5 independent cultures). **i** Schematic of 6-azauridine targeting pyrimidine biosynthesis created with Biorender.com (Agreement number KY26CFPJC3). **j** Relative percentage of CD71/GlyA$^{high}$ K562 cells treated with 2.5 µM SHIN1, 5 µM 6-azauridine with or without 100 µM uridine (URI) (n = 4 independent cultures). **k** Immunoblot analysis of RPS6 phosphorylation in K562 cells treated as in (**j**) for the indicated times (representative of three independent experiments). **l** Relative percentage of CD71/GlyA$^{high}$ K562 CTR and TSC2 KO cells treated with 5 µM 6-azauridine (n = 4 independent cultures). Data are shown as the mean ± s.e.m. P-values were calculated with a repeated measure one-way ANOVA with Dunnett's multiple comparisons test (**b, d, f, j**), two-way ANOVA with Sidak's multiple comparisons test (**h**) or two-tailed paired t-test (**l**). Exact p-value in (**j**) comparing UT with 6-AZA treated cells is 0.000037. Source data are provided as a Source Data file.

---

Fig. 7a). Additionally, it resulted in decreased abundance of m + 3 glycine isotopologue (Fig. 5a). Of note, imatinib did not alter the serine and glycine labelling pattern, suggesting that SHMT activity was maintained (Fig. 5a and Supplementary Fig. 7a). Through generation of glycine, folate metabolism contributes to glutathione (GSH) production, the most abundant antioxidant in the cell. Intriguingly, serine incorporation into GSH and its oxidised form (GSSG) persisted in TKI-treated K562 cells, while SHIN1 completely abrogated the contribution of extracellular serine to GSH and GSSG (Fig. 5b and Supplementary Fig. 7b). It should be mentioned that imatinib, similarly to SHIN1, reduced serine labelling into the purine nucleotide pool (Fig. 5c and Supplementary Fig. 7c). Furthermore, serine contribution to glycine, GSH and GSSG was maintained in imatinib-treated primary CML CD34$^+$ cells, while SHIN1, and consequently, combination of SHIN1 with imatinib, significantly reduced levels of relevant labelled fractions in GSH and GSSG (Fig. 5d, e and Supplementary Fig. 7d). These data suggest that TKI therapy does not affect the incorporation of serine into GSH, implying a sustained ability for TKI treated CML cells to maintain redox defence.

Given the antiproliferative effect observed following mitochondrial folate metabolism inhibition in K562 cells and in vivo, we labelled CML CD34$^+$ cells with a fluorescent tracer of cell division (CTV) and treated them with imatinib, SHIN1 or a combination of both. Simultaneously, we supplemented SHIN1-treated cells with formate, to assess whether we could recapitulate the findings obtained in K562 cells. SHIN1, alone or in combination with imatinib, impaired the proliferation of primary CML CD34$^+$ cells, even to a greater extent than imatinib alone (Fig. 6a and Supplementary Fig. 6a). Formate supplementation restored proliferation in SHIN1 treated cells (Fig. 6a and Supplementary Fig. 6a). Furthermore, SHIN1 and imatinib single treatments reduced the colony formation potential of patient-derived CD34$^+$ cells (n = 7) in colony-forming cell (CFC) assay (Fig. 6b and Supplementary Fig. 6b). Moreover, SHIN1 sensitised CD34$^+$ cells to TKI treatment. Formate was sufficient to reverse the effect of SHIN1 on colony formation potential of primitive CML cells (Fig. 6b and Supplementary Fig. 6b). In contrast, neither single agent nor combination treatment affected the colony forming potential of normal CD34$^+$ cells (Fig. 6c).

We next hypothesised that SHIN1 primes a fraction of primitive CML cells out of quiescence, thus we treated CML CD34$^+$ cells (n = 4) with imatinib, SHIN1 or a combination of both for 48 h and stained them with the proliferation marker Ki67, and the DNA stain DAPI (Supplementary Fig. 8c). SHIN1, alone or in combination with imatinib, increased the fraction of cells at the S-G$_2$-M phase (Supplementary Fig. 8d). Furthermore, we observed a noticeable, but not statistically significant, decrease in the fraction of cells at the G$_0$ phase (Supplementary Fig. 8d). To further examine the clinical relevance of our findings, we utilised a robust CML xenotransplantation model. Since SHIN1 has modest efficacy in vivo due to drug clearance[33], we treated patient-derived CML CD34$^+$ cells as depicted above, and transplanted

them into sub-lethally irradiated NRGW mice (Fig. 6d). After 12 weeks, bone marrow cells were collected, and engraftment determined by flow cytometry (Supplementary Fig. 8e). We observed a reduction in the percentage of human CD45$^+$ leucocytes and total number of human bone marrow CD45$^+$ cells in the SHIN1 and imatinib combination group when compared to imatinib monotherapy, although this did not reach statistical significance (Fig. 6e and Supplementary Fig. 8f). Similar results were obtained for the total and relative number of human CD45$^+$CD34$^+$ cells (Fig. 6f and Supplementary Fig. 8g). However, when examining the most primitive population of human LSCs (CD45$^+$CD34$^+$CD38$^-$), combination of SHIN1 with imatinib conferred a 75% reduction of the absolute number of LSCs compared to imatinib alone (Fig. 6g). A significant decrease in the relative number of human LSCs was also observed (Supplementary Fig. 8h). Notably imatinib increased the total number of LSCs, supporting literature that suggests that TKI therapy enriches for the most primitive population of LSCs (Fig. 6g). Together these data highlight that 1C metabolism inhibition sensitises therapy resistant LSCs to targeted therapy.

Lastly, despite previous reports of limited stability of SHIN1 in vivo[33], Pikman et al.[35] demonstrated that SHIN1 decreases the leukaemic burden in the bone marrow and spleen of mice transplanted with T-ALL cells. To test the effect of SHIN1, as a single agent or in combination with imatinib in vivo, we transplanted luciferase expressing KCL22 cells into non-irradiated NRGW mice. After 13 days, mice were randomised into four groups and treated with vehicle, imatinib, SHIN1 and SHIN1 combined with imatinib for 21 days. Combination of SHIN1 with imatinib significantly reduced the tumour burden compared to vehicle or imatinib treated mice, as measured by in vivo bioimaging following 14 days of treatment (Fig. 6h, i). This significantly prolonged tumour-free survival in relation to mice treated with imatinib monotherapy (Fig. 6j). Imatinib did not reduce tumour burden or prolong survival of mice (Fig. 6h–j). As KCL22 are blast phase CML cells, the inability of imatinib to suppress tumour formation is in line with the poor in vivo efficacy of the drug in the blast phase of the disease.

## Discussion

Our findings demonstrate an undescribed role for mitochondrial folate metabolism in CML, a malignancy with a well-established cancer stem cell hierarchy. Previous work from our lab highlighted the importance of mitochondrial metabolism, and particularly OXPHOS, in therapy resistant LSCs[29]. Furthermore, it has been reported that increased serine catabolism to formate is a hallmark of oxidative cancers[8]. However, while targeting folate metabolism is an active area of research in cancer, the role of 1C metabolism in CML LSCs is unexplored. We show that folate metabolism is deregulated in human stem and progenitor cells. This deregulation is mainly driven by the differential expression of three mitochondrial-specific enzymes, ALDH1L2, SHMT2 and MTHFD2. Moreover, we found that mitochondrial folate

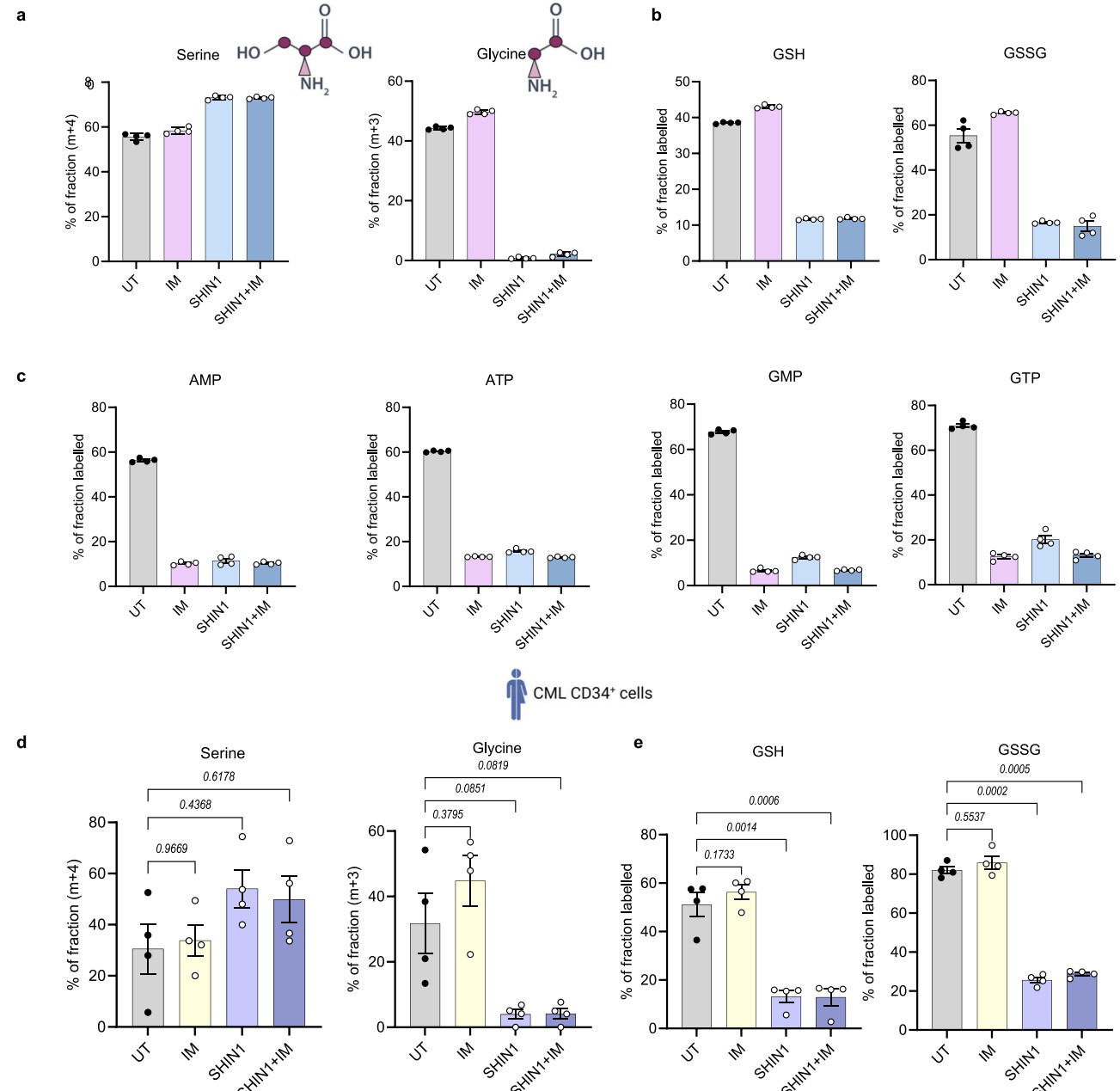

**Fig. 5 | Imatinib maintains serine contribution to glutathione.** Fractional labelling of serine, glycine (**a**), GSH, GSSG (**b**), and purine nucleotides (AMP, ATP, GMP, GTP) (**c**) from K562 cells treated for 24 h with 2 μM imatinib (IM), 2.5 μM SHIN1, or a combination of both in medium containing 140 μM $^{13}C_3$$^{15}N_1$-serine. ($n = 4$ independent wells from individual experiment). Schematics for serine and glycine structures were created with Biorender.com (Agreement numbers: EZ26CFMM6G and WI26CFWJC3 respectively). Fractional labelling of serine, glycine (**d**) and GSH, GSSG (**e**) from CML CD34$^+$ cells treated for 24 h with 2 μM imatinib (IM), 2.5 μM SHIN1, or a combination of both in medium containing 140 μM $^{13}C_3$$^{15}N_1$-serine ($n = 4$ patient samples). Fractional labelling represents fraction of the sum of all labelled isotopologues for each metabolite. Schematic for patient cells created with Biorender.com (Agreement number SH26CFO8OE). Data are shown as the mean ± s.e.m. *P*-values were calculated with a repeated measure one-way ANOVA with Dunnett's multiple comparisons test (**d**, **e**). Source data are provided as a Source Data file.

metabolism, primarily through its contribution to purine biosynthesis, is essential for CML cell proliferation and tumour formation in mice. Specifically, we demonstrate that mitochondrial 1C metabolism is indispensable for purine synthesis and genetic ablation of SHMT2 inhibits tumour formation capacity of CML cells. This is in contrast to work from ref. 5. which showed that HCT116 colon cancer cells, engineered with mitochondrial 1C enzyme deletions, form xenograft tumours by generating cytosolic 1C units from serine via the enzyme SHMT1. Furthermore, two independent studies have described that inhibition of both mitochondrial and cytosolic 1C metabolism

branches is necessary to halt proliferation of T-ALL cells[35,49]. Thus, we hypothesise that the dependency on mitochondrial 1C metabolism might be a unique feature of CML or myeloid leukaemias.

Additionally, our work demonstrates that depletion of 1C units and purine nucleotides results in differentiation of CML cells and patient-derived progenitor cells. We show that the differentiation is not a consequence of increased ROS, one of the main drivers of differentiation in HSCs. We also conclude that initiation of differentiation does not depend on AMPK activity but is linked to purine sensing through mTORC1 signalling. Recent work has described that

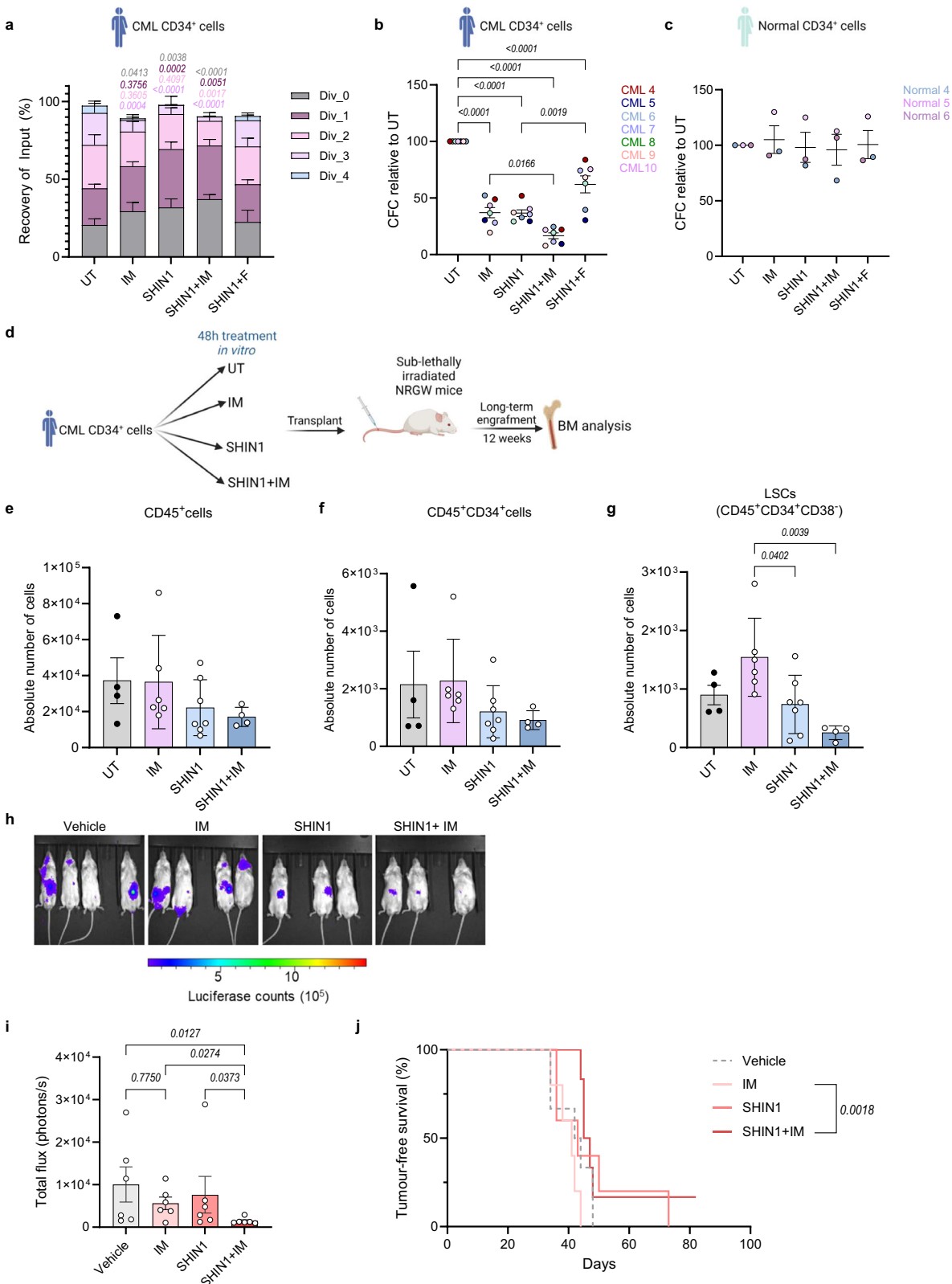

impairment of purine biosynthesis leads to inactivation of mTORC1[47,48]. Similarly, Wilke and colleagues[50] demonstrated that loss of SHMT2 activity leads to mTORC1 inhibition and autophagy induction, disrupting the survival of Burkitt lymphoma cells. Furthermore, it has been reported that mTORC1 promotes serine catabolism to formate and purine biosynthesis through induction of the activating transcription factor 4 (ATF4)[51], supporting the existence of a positive

feedback loop between folate metabolism, de novo purine synthesis and mTORC1 signalling. Here, we observe that while pharmacological inhibition of 1C metabolism leads to suppression of mTORC1 activity, this can be rescued either by reconstitution of 1C units or purine nucleotides, preferably adenine. These findings are consistent with the study of ref. 47. which demonstrated that mTORC1 acutely senses changes in adenylates rather than guanylates. Furthermore, our data

**Fig. 6 | SHIN1 selectively targets LSCs in combination with standard CML therapy. a** CellTrace Violet (CTV) tracked proliferation of CML CD34+ cells following 72 h of treatment with 2 μM IM, 2.5 μM SHIN1 with or without the addition of 1 mM F, or a combination of imatinib and SHIN1, reported as recovery of input (%), where input represents number of input cells (*n* = 4 patient samples). Relative colony numbers of CML CD34+ (**b**) (*n* = 7 patient samples) or normal CD34+ (**c**) (*n* = 3 patient samples) cells following treatment as in (**a**). Schematics for CML and normal patient samples were created with Biorender.com (Agreement numbers: SH26CFO8OE and IK26CFP2TJ, respectively) **d** Schematic overview of PDX experimental design: CML CD34+ cells were treated with 2 μM imatinib, 2.5 μM SHIN1, or a combination of both for 48 h, and transplanted into sub-lethally irradiated female NRGW41 mice. After 12 weeks, bone marrow was collected, and engraftment determined by flow cytometry. Created with Biorender.com (Agreement number: ZP26CG12RD). Number of human CD45+ (**e**), CD34+ (**f**), and CD34+CD38− (**g**) CML cells in bone marrow at experimental endpoint (*n* = 4 mice for untreated group; *n* = 6 mice for imatinib-treated group; *n* = 7 mice for SHIN1-treated group; *n* = 4 mice for combination group). Representative images from treated mice with vehicle, imatinib (50 mg/kg, BID), SHIN1 (100 mg/kg) or a combination of both (**h**) and quantification (**i**) of changes in tumour burden as assessed by bioimaging in NRGW41 mice transplanted with luciferase expressing KCL22 cells following 14 days of treatment (*n* = 6 mice per group). **j** Tumour-free survival of mice subjected to treatment is depicted in (**h**) (*n* = 6 mice per group). Data are shown as the mean ± s.e.m. *P*-values are derived from two-way ANOVA with Tukey's multiple comparisons test (**a**) relative to untreated CML CD34+ cells, ordinary one-way ANOVA with Dunnett's multiple comparisons test (**b**, **g**, **i**) and log-rank (Mantel–Cox) test for survival analysis (**j**). Exact *p*-values in (**a**) comparing division 3 between UT and SHIN1 treated cells is 0.000052, division 3 between UT and SHIN1 + IM is 0.000001, division 0 between UT and SHIN1 + IM is 0.000007. Source data are provided as a Source Data file.

support that reactivation of mTORC1 correlates with suppression of differentiation. Ultimately, we found that purine depletion fails to suppress mTORC1 activity and induce erythroid maturation in TSC2-deficient cells, supporting that purine sensing through the mTORC1 network mediates differentiation of CML cells. In contrast, SHIN1 inhibited mTORC1 signalling and increased expression of the monocytic marker CD11b in TSC2 KO AML cells, implying that purine sensing does not rely on the TSC complex in this setting. While rapamycin inactivated mTORC1, it failed to promote differentiation in AML cells. As a result, delineating the interplay between purine sensing, mTORC1 activity and differentiation in AML cells requires further investigation. Additionally, we described that pyrimidine depletion initiates differentiation of CML cells, while it only modestly affects mTORC1 activity. These data suggest that impairment of pyrimidine biosynthesis induces differentiation through a separate mechanism, independently of mTORC1 signalling. Indeed, in agreement with our data, it has been recently suggested that pyrimidine depletion alters cell fate of AML cells through DNA replication stress[52].

In terms of clinical relevance, we found that folate metabolism inhibition synergises with TKI treatment to target therapy insensitive LSCs. Resistance of LSCs to standard therapy has been associated with their undifferentiated and quiescent state[53,54]. Thus, we propose that LSCs upregulate folate metabolism to drive leukaemogenesis, without significantly deregulating the normal differentiation process. Intriguingly, Dohla et al.[55] uncovered that in stem-like human mammary epithelial cells, purine synthesis was upregulated in daughter cells that were destined to maintain stemness[56–58]. In relation to CML, differentiation therapy, which would force LSCs to terminally differentiate and have a finite lifespan, has gained interest and is considered a promising therapeutical approach[58–63]. However, these approaches have not shown a strong therapeutic benefit for CML patients in clinical trials, highlighting the need for the development of more efficient differentiation inducers[56]. We suggest that inhibition of folate metabolism through impairment of purine biosynthesis can prime LSCs out of quiescence and thereby sensitise them to TKI therapy, which preferentially targets more differentiated cells. Although more work is required to confirm this concept, we found that inhibition of 1C metabolism alone or in combination with TKI therapy increased the fraction of CML CD34+ cells found in the S-G$_2$-M phase, with a hint of decrease in the fraction of quiescent (G$_0$) cells. Notably, TKI therapy was found to maintain serine incorporation to GSH and GSSG, suggesting that the synergistic effect with folate metabolism inhibition might be related to impeded antioxidant defence. Serine metabolism contributes to the maintenance of the GSH pool in two ways. Firstly, serine-derived glycine combines with cysteine and glutamate to form GSH. Secondly, serine catabolism contributes to generation of mitochondrial NADPH[64,65], which is consumed during the reduction of GSSG to GSH. Whether alterations of the GSH pool sensitises cells to TKI therapy requires further investigation.

Lastly, it has been shown that suppression of MTHFD2 leads to differentiation of AML cells. Pikman et al.[66] demonstrated that formate supplementation did not restore proliferation following suppression of MTHFD2, implying that the antiproliferative effect of MTHFD2 ablation was not a result of 1C unit depletion. In the present study, we describe that folate metabolism inhibition initiates differentiation of AML cells, and this can be reversed by addition of formate or purine nucleotides. Moreover, together these data indicate that mitochondrial 1C metabolism is a targetable vulnerability for myeloid leukaemias and suggest that folate metabolism-driven differentiation may have a broader application to related malignancies of stem cell origin. Given that *SHMT2* and *MTHFD2* are overexpressed in cancer, mitochondrial 1C metabolism inhibitors might be more selective than cytosolic antifolates like methotrexate. Methotrexate inhibits the activity of dihydrofolate reductase (DHFR), but has affinity for folate-dependent enzymes such as thymidylate synthase (TYMS) and 5-aminoimidazole-4-carboxyamide ribonucleotide formyltransferase[67,68]. It has been shown that DHFR and TYMS are expressed in many proliferating tissues, in which the adverse effects of MTX are observed[7]. Our findings raise the possibility that inhibiting mitochondrial 1C metabolism may be more selective and have less adverse side effects.

## Methods

### Ethical approval

Patients gave informed consent in accordance with the Declaration of Helsinki and with approval of the National Health Service Greater Glasgow Institutional Review Board and Clyde Institutional Review Board. Ethical approval has been granted by the West of Scotland Research Ethics Service (REC reference15:/WS/0077 and 10/S0704/60).

Animal experiments were performed according to all applicable laws and regulations with ethical approval from the University of Glasgow Animal Welfare and Ethical Review Board (AWERB) under approved project licence PPL PP2518370 and personal licence PIL I1F599357.

### Patient derived samples

Primary CML samples were obtained from individuals in chronic phase CML at the time of diagnosis and were product of leukapheresis. Equal number of female (7) and male (7) patient samples were used. Patient ages ranged from 28–70 years old, with the median age at diagnosis being 49.5 years old. Patient information is available in Supplementary Table 1. Normal samples were (i) surplus cells collected from femoral-head bone marrow, surgically removed from patients undergoing hip replacement or (ii) leukapheresis products from individuals with non-myeloid Philadelphia negative haematological disorders. CD34+ cells were isolated using the CD34 MicroBead Kit or CliniMACS (catalogue no. 130-100-454; Miltenyi Biotec).

## Cell culture

CML and AML cell lines were cultured in RPMI 1640 medium (catalogue no. 11875093; Thermo Fisher Scientific) supplemented with 100 IU/mL penicillin/streptomycin, 2 mM L-glutamine (catalogue no. A2916801; Thermo Fisher Scientific), and 10% (v/v) fetal bovine serum (FBS, catalogue no. 10100147). HCT116 cells were cultured in DMEM medium (catalogue no. 21969035; Thermo Fisher Scientific) supplemented as stated above. SHMT2 KO cells were cultured with 1% hypoxanthine/thymidine supplement (HT, catalogue no. 11067030; Thermo Fisher Scientific). HT supplement was removed prior to any experimental procedures. Primary samples were cultured in serum-free medium (SFM) consisting of IMDM (catalogue no. 12440061; Thermo Fisher Scientific) and supplemented with 20% bovine serum albumin (BSA), insulin, and transferrin (BIT, catalogue no. 09500; STEMCELL Technologies), 40 µg/ml low-density lipoprotein (catalogue no. L4646; Sigma-Aldrich), 0.1 mM 2-mercaptoethanol (catalogue no. 31350-010; Thermo Fisher Scientific) and 100 IU/mL penicillin-streptomycin (catalogue no. 15140122; Thermo Fisher Scientific). Cells were also supplemented with a physiological growth factor cocktail consisting of 0.2 ng/ml stem cell factor (SCF, catalogue no. 573902; BioLegend), granulocyte macrophage colony-stimulating factor (GM-CSF, catalogue no. 578602; BioLegend), 1 ng/ml interleukin-6 (IL6, catalogue no. 570802; BioLegend), macrophage inflammatory protein (MIPα, catalogue no. 300-08; PeproTech) and 0.05 ng/ml leukaemia inhibitory factor (LIF, catalogue no. 300-05; PeproTech). Both cell lines and primary cells were maintained at 5% $CO_2$ and 37 °C. All cell lines were obtained from DSMZ and regularly tested to ensure they were free of contamination such as mycoplasma. Cell numbers were monitored using the CASY automated cell counter (Roche).

Where indicated, the following compounds were added to the medium: 2.5 µM SHIN1 (catalogue no. HY-112066; Cambridge Biosciences), 1 mM AICAR (catalogue no. A8184-APE; ApexBio), 10 nM rapamycin (catalogue no. R-5000; LC Laboratories), 5 µM 6-azauridine (catalogue no. A1888; Sigma-Aldrich), 5 µM HCQ (catalogue no. 10646271; Fischer Scientific), 3 µM MRT403 (LifeArc), and 2 µM imatinib mesylate (catalogue no. I-5508; LC Laboratories). For formate supplementation experiments, sodium formate (catalogue no. 141-53-7; Sigma-Aldrich) was added at a concentration of 1 mM. Hypoxanthine (catalogue no. A11481; Thermo Fisher Scientific) and thymidine (catalogue no. A11493.06; Thermo Fisher Scientific), were used in a concentration of 100 µM and 16 µM, respectively. For purine nucleotide supplementation, cells were cultured in 30 µM adenine (catalogue no. A2786; Sigma-Aldrich) or/and 30 µM guanine (catalogue no. G6779; Sigma-Aldrich). For pyrimidine rescue experiments, uridine (A15227; Thermo Fisher Scientific) was added at a concentration of 100 µM.

## Growth curves and proliferation rates

K562 cells were seeded at a density of 50,000 cells/ml in a 12-well plate. Indicated treatments were added to separate wells (three wells per condition). Cell counts were obtained at 24 h intervals using the CASY automated cell counter (Roche). The proliferation rate was determined based on the following formula:

$$\text{Proliferation rate(doublings per day)} = \log_2(\text{final cell count}/\text{initial cell count})/\text{no.of days.}$$

## Cell cycle analysis

K562 and primary CML cells were cultured at 100,000 cells/ml in the presence of the indicated treatments. Following a 72 h (h) incubation, cells were fixed in cold 70% ethanol while vortexed and then incubated at −20 °C for at least 2 h. K562 cells were then stained (100 µl/test) with propidium iodide (PI, catalogue no. P1304MP; Thermo Fisher Scientific; 5 µl/test) in the presence of 100 µg/ml of RNase A (QIAGEN), while primary samples were stained Ki67-Alexa 488 (catalogue no. 151204;

clone 11F6; RRID: AB_2566800; Biolegend; 0.25 µg/test) for 20 min. Primary CML samples were then stained with DAPI (catalogue no. 422801; Biolegend; 3µM). Samples were then analysed using the FaCSVerse flow cytometer (BD Biosciences). Analysis of cell cycle stages was performed with FlowJo V10.7.

## Generation of knockout and stable cell lines

To generate CRISPR–Cas9-mediated gene knockouts in CML and AML cells, we used guide sequences as described in Supplementary Table 2. Synthesis of selected guides was conducted by Integrated DNA Technologies. Oligonucleotides were annealed and cloned in BsmBI–digested lentiCRISPRv.2-puro plasmid (catalogue no. 52961; Addgene). Lentiviral particles for cloned lentiCRISPRv.2 were generated by transfecting human embryonic kidney (HEK) 293FT cells with pCMV-VSV-G envelope and psPAX2 packaging vectors using the calcium phosphate method[69]. Viral supernatant was harvested after 48 h and transferred onto target CML cells. Cells were then selected using 3 µM/ml puromycin and presence of guides was validated by immunoblotting. K562 ULK1 KO and K562 ATG7 KO cells were previously generated as described by ref. [25].

## Western blot analysis

Cells from a six-well plate were rinsed with PBS and lysed into RIPA buffer (catalogue no. 89900; Thermo Fisher Scientific) containing cOmplete Protease (Roche) and PhosSTOP phosphatase (Roche) inhibitors. Cell lysates were cleared by centrifugation at 16,000 g for 15 min at 4 °C. Protein concentration was quantified by bicinchoninic acid assay (Thermo Fisher Scientific). Lysates were resolved in 4 to 12% SDS–polyacrylamide gel electrophoresis gels and transferred to PVDF membranes. Blocking was performed for 1 h in 2% BSA (in Tris-buffered saline, 0.01% Tween (TBS-T)) followed by incubation overnight with indicated primary antibodies (Supplementary Table 3) at 4 °C. The next day, membranes were incubated with the appropriate horse-radish peroxidase conjugated secondary antibodies (Supplementary Table 3) at room temperature for 1 h. Bands were detected by ECL reagent (catalogue no. RPN2235; Cytiva) and developed using the Odyssey FC imaging system V.5.2.

## Erythroid maturation analysis

To evaluate erythroid differentiation expression of cell surface markers GlyA and CD71 were assessed. K562 cells were cultured at 100,000 cells/ml in the presence of the indicated treatments. CML CD34$^+$ cells were cultured at 100,000 cells/ml in SFM medium with the addition of EPO (3 U/ml, catalogue no. 100-64; PeproTech). Following a 72 h incubation, cells were stained (100 µl/test) with fluorophore-conjugated antibodies GlyA-PE (catalogue no. 349106; clone HI264; RRID: AB_10640739; Biolegend; 1 µl) and CD71-APC (catalogue no. 551374; clone M-A712; RRID: AB_398500; BD Biosciences; 2 µl) for 20 min and analysed using the BD FacsVerse flow cytometer and FlowJo V.10.8.1.

## Myeloid maturation analysis

Expression of the monocytic cell surface marker CD11b was assessed in AML cell lines. Following 72 h incubation with the indicated treatments, THP1 and MOLM13 were stained (100 µl/test) with CD11b-PE (catalogue no. 301306; clone ICRF44; RRID: AB_314158; Biolegend; 0.5 µl) for 20 min and analysed by flow cytometry.

## Mitochondrial ROS measurements

K562 cells (25,000 cells/ml) were treated with the indicated treatments for 72 h and stained with 5 µM MITOSOX (catalogue no. M36008; Thermo Fisher Scientific) in 1 ml of phosphate buffered saline (PBS) solution. Cells were then incubated for 30 min at 37 °C and analysed by flow cytometry.

## Cellular division tracking

CML CD34$^+$ cells were stained with 1 μM CellTrace Violet (catalogue no. C34557; Thermo Fisher Scientific) for 30 min at 37 °C. The reaction was quenched by adding cell culture medium containing 10% FBS. Cells were then suspended in SFM and treated with the indicated treatments. After 72 h, CellTrace Violet staining was assessed by the BD FACSVerse flow cytometer. Analysis was conducted using FloJo V.10.8.1. Each division was gated using the geometric mean fluorescence intensity (MFI) to determine the divisions, with each subsequent division having half the fluorescent intensity of the previous division. Data were when normalised to cell counts collected after 72 h and reported as recovery of input (%), relative to the number of input cells.

## Steady-state metabolic profiling

For steady-state metabolite measurements, CML cells were seeded in different experimental conditions in standard RPM1 1460 medium one day prior to metabolite extraction. On the day of the metabolite extraction, cells were counted using the CASY automated cell counter. Cell number was multiplied by cell size (peak volume) and this number was used to calculate the volume of extraction solvent. Cells were washed twice with ice-cold PBS with pulse centrifugation (12,000 g, 15 s, 4 °C) before a 5 min incubation step with extraction buffer (50:30:20, v/v/v acetonitrile/methanol/water). Cells were then centrifuged at 16,000 g for 10 min at 4 °C, and the supernatant was transferred to liquid chromatography-mass spectrometry (LC-MS) glass vials and kept at −80 °C until measurements were performed.

Samples were analysed using a Thermo Ultimate 3000 High pressure liquid chromatography (HPLC) system (Thermo Scientific) coupled to a Q Exactive Orbitrap mass spectrometer (Thermo Scientific). The HPLC system consisted of a ZIC-pHILIC column (SeQuant, 150 × 2.1 mm, 5 μm, Merck KGaA), with a ZIC-pHILIC guard column (SeQuant, 20 × 2.1 mm) and an initial mobile phase of 20% 20 mM ammonium carbonate, pH 9.4, and 80% acetonitrile. For the analysis, 5 μl of metabolite extracts were injected into the system, and metabolites were separated over a 15 min mobile phase gradient decreasing the acetonitrile content to 20%, at a flow rate of 200 μl/min and a column temperature of 45 °C. Metabolites were then detected using the mass spectrometer operating in polarity switching mode. All metabolites were detected across a mass range of 75–1000 m/z at a resolution of 35,000 (at 200 m/z), and with a mass accuracy below 5 ppm. Data were acquired using Thermo Xcalibur software. The peak areas of different metabolites were determined using the TraceFinder 4.1 software (Thermo Fischer Scientific). Metabolites were identified by accurate mass of the singly charged ion and by known retention times on the pHILIC column.

## Stable isotope enrichment analysis using $^{13}C_3 {}^{15}N_1$-serine

For tracing experiments, CML, primary CML, and HCT116 cells were cultured in Plasmax, a physiological cell culture medium[70], containing 140 μM $^{13}C_3 {}^{15}N_1$-serine for 24 h (catalogue no. CNLM-474-H-PK; Cambridge Isotope Laboratories). CML and HCT116 cells were cultured in Plasmax supplemented with 10% dialysed FBS (catalogue no. A3382001; Thermo Fisher Scientific). Primary CML cells were cultured in Plasmax supplemented with physiological growth factors as described above (see section Cell culture), however instead of BIT, medium was supplemented with 1 mg/ml Albumax II (catalogue no. 11021037; Thermo Fisher Scientific), 10 μg/ml insulin (catalogue no. I9278; Sigma-Aldrich), and 7.5 μg/mL transferrin (catalogue no. T4132; Sigma- Aldrich). Cell culture, harvest, metabolite extraction, analysis by LC-MS was performed as described for steady state metabolite profiling (see above). The $^{13}C$ $^{15}N$ labelling pattern was determined by quantifying peak areas for the accurate mass of all isotopologues of each metabolite. Labelled isotopologues were defined as mass + n (m + n), where n = number of $^{13}C$ or $^{15}N$ incorporated. Fig.s include clarification whether (m + n) consists of only $^{13}C$ isotopologues, or a combination of $^{13}C$ or $^{15}N$. We calculated the fractional enrichment, which describes the relative contribution of stable isotopes to the total area of each compound.

## Stable isotope enrichment analysis using 2,3,3-$^2H_3$-Serine

CML and HCT116 were cultured in plasmax containing 140 μM 2,3,3-$^2H_3$-serine for 24 h (catalogue no. 105591-10-4; Cambridge Isotope laboratories). On the day harvested and extracted as described above (see section Steady state metabolite profiling). Next, cell extractes (250 μL) were thoroughly mixed with 5 μl of 25 μg/ml internal standard disodium deoxythymidine 5′-monophosphate-$^{13}C_{10}$,$^{15}N_2$ (catalogue no. 648590; Merck). The samples were dried under $N_2$ at room temperature using a 36-port Stuart drying manifold SBH130D/3 (Cole Parmer). The dried extracts were then resuspended in 25 μL extraction solution (50:30:20, v/v/v acetonitrile/methanol/water). Samples were analysed by LC-MS as described above. Mass spectrometric detection was performed in positive ionization full scan mode across a mass range of 310–500 m/z at a resolution of 140,000 at 200 m/z. Minimum automatic gain control (AGC) was set at 3e$^6$ and maximum injection time was set at 220 ms. Peaks were annotated by matching accurate masses, adducts and retention times to reference standards. MS data were analysed using Skyline software (version 21.2.0.369). In Skyline, instrument settings were set to orbitrap, and the resolution was set at 140,000. A transition list containing the compound names, accurate masses, explicit retention times with 2 min retention time window, and polarity. Extracted ion chromatograms were generated for each compound and chromatographic peaks were carefully inspected and integrated. Relative quantification between sample groups was performed using area ratios to internal standard. Labelled isotopologues were defined as mass + n (m + n), where n = number of $^2H$ incorporated. We calculated the fractional enrichment, which describes the relative contribution of stable isotopes to the total area of each compound.

## Quantitative analysis of extracellular metabolites

Quantity of extracellular glucose and lactate was determined using the YSI 2900 biochemistry analyser as per manufacturer instructions. The concentration of each metabolite was normalised to cell number and the rate of uptake or secretion per hour was calculated relative to medium only sample. The exchange rate for each metabolite was calculated using the formula below:

$$Exchange\ rate = \Delta metabolite / ((cell\ number\ day\ 0 + cell\ number\ day1)/2),$$

where Δmetabolite = ((x) mmol spent medium - (x) mmol medium only sample)

## GC-MS formate analysis

Formate quantification using gas chromatography-mass spectrometry (GC-MS) was performed as previously described by ref. 34. Media samples from CML CD34$^+$ cells (40 μL) were added in 20 μl of 50 μM internal standard sodium $^{13}C$,$^2H$-formate (m + 2), 10 μl of 1 M sodium hydroxide, 5 μl of benzyl alcohol and 50 μl of pyridine. While vortexing, derivatisation of formate was commenced by the addition of 20 μl of methyl chloroformate. 200 μl of water and 100 μl of methyl tertiary butyl ether were then added, and the samples were vortexed for 20 s and centrifuged at 10,000 g for 5 min. The resulting apolar phase containing the formate derivative (benzyl formate) was transferred to a GC-vial and capped. Formate standards and blank samples (water) were prepared in the same manner and analysed with the experimental samples. Derivatized formate samples were analysed with an Agilent 7890B GC system coupled to a 7000 triple quadrupole MS system[34]. MassHunter Quantitative analysis software (Agilent Technologies) was used to extract and process the peak areas for m + 0, m + 1, m + 2 formate. After correction for background signals, quantification was

performed by comparing the peak area of formate (m/z of 136) and of m + 1 formate (m/z of 137) against that of m + 2-formate (m/z of 138).

## Measurement of extracellular acidification rate

ECAR of CML cells was measured using the Seahorse XF96 flux analyser (Seahorse Bioscience). The XF96 well plate was coated with 22.6 µg/ml Cell-Tak solution (catalogue number. 354240; Corning) and left for at least 30 min at room temperature. Following a 24 h incubation with indicated treatments, K562 cells were suspended in XF Assay Medium (catalogue no. 100965–000; Seahorse Bioscience) supplemented with 2 mM glutamine and seeded at a concentration of 70,000–100,000 cells/well in the pre-coated XF96 well plate. Measurement of ECAR was performed at baseline and after sequential injections of glucose (11 mM), oligomycin (0.7 µM), and 2-deoxyglucose (5 mM). Glycolysis and glycolytic capacity relative to untreated or vector control were calculated according to manufacturer's instructions.

## Computational analysis

Affymetrix microarray dataset (E-MTAB-2581) was analysed using Bioconductor. Packages *oligo* and *arrayQualityMetrics* were used for quality control and pre-processing, while *limma* was used for differential gene expression analysis. Statistical significance was determined using the Benjamini–Hochberg method. Adjusted $p < 0.05$ and $\log_2 FC \geq abs (0.5)$ were considered to be significant. GSEA (version 4.1.0) analysis was conducted on pre-ranked lists (ranked by pi score calculated by multiplying $\log_2$ FC by $-\log_{10}$ (adjusted $p$-value)).

## Colony-forming cell assay

Primary CD34+ cells were cultured in SFM in the presence of indicated treatments. After 72 h, 3000 cells per condition were seeded to methylcellulose based medium (catalogue no. 04034; STEMCELL Technologies) in duplicate, and colonies were counted after 14 days.

## Mouse work

Animals were housed in a pathogen-free facility at the Cancer Research UK Scotland Institute and kept in day/night cycles (12 h each), with temperature of 20–24 °C and humidity 45–65%. Feeding and water were ad libitum.

## Tumour xenograft growth

KCL22 control or SHMT2 KO cells (4,000,000 cells per mouse) were transplanted via tail vein injection into female NRGW[41] mice aged 8–12 weeks. Mice were euthanized according to institutional guidelines before reaching the threshold maximum tumour burden. Tumour size was recorded using a calibre, and tumour burden was calculated using the formula (volume = $0.5 \times$ (length $\times$ width$^2$)).

To test the efficacy of SHIN1 in vivo, KCL22 luciferase-expressing cells were transplanted via tail vein injection into NRGW[41] mice aged 8–12 weeks. Tumour burden was assessed weekly using bioluminescence imaging by injecting mice subcutaneously with 200 µL L d-Luciferin (catalogue no. 122799-2; PerkinElmer), anaesthetising them with 2–3% isoflurane, and imaging them on an IVIS Spectrum (PerkinElmer). After 13 days, mice were separated into 4 cohorts (vehicle, imatinib, SHIN1, SHIN1 with imatinib) and treated for 21 days. Imatinib was administered BID at 50 mg/kg (oral gavage), whereas SHIN1 was administered at 100 mg/kg (intraperitoneal injection).

## Mouse engraftment using patient-derived xenografts

To study the in vivo engraftment of patient derived CML CD34+ cells, CML CD34+ cells were cultured in SFM in the presence with appropriate treatment for 48 h. Cells (1,000,000 cells per mouse) were then harvested, washed and transplanted into sub-lethally irradiated (2.0 Gy) female NRGW[41] mice aged 8–12 weeks. Following 12 weeks, bone marrow was extracted from the hip, tibia, and femur of each mouse. Cells were stained (100 µl/test) with anti-mouse CD45-APC-Cy7 (catalogue no. 557659; clone 30-F11; RRID: AB_396774; BD Biosciences; 0.5 µl), anti–human CD45-FITC (catalogue no. 555482; clone HI30; RRID: AB_395874; BD Biosciences; 10 µl), anti–human CD34-APC (catalogue no. 555824; clone 581; RRID: AB_398614; BD Biosciences; 1 µl) and anti–human CD38-PerCP (catalogue no.303520; clone HIT2; RRID: AB_893313; Biolegend; 1 µl) antibodies and analysed by flow cytometry as described above.

## Statistics and reproducibility

Statistical analyses were performed using Prism 9 (Graphpad) software. The number of biological replicates and applied statistical analysis are indicated in the Fig. legends. For in vitro experiments, no statistical methods were used to predetermine sample size. Sample sizes were estimated according to common practice for each experimental design, preliminary and previous experiments to estimate variability. In vitro experiments were repeated at least on three separate occasions (independent cultures), except where noted. For experiments with CML primary cells, a minimum of four samples was used to give adequate power. Cells were randomly plated/treated/analysed during each experiment for all in vitro experiments. Investigators were not blinded during data processing and analysis of in vitro data. Statistical significance for parametric data was determined by paired or unpaired two-tailed *t*-tests. One-way analysis of variance (ANOVA) was used to assess statistical significance between three or more groups with one experimental parameter, while two-way ANOVA was used to determine significance between multiple groups with two experimental parameters. Nonparametric data were analysed using two-sided Mann–Whitney tests. Kruskal–Wallis test was used to assess statistical significance between three or more groups with one experimental parameter, when the data were not normally distributed. For in vivo work, while no statistical methods were used to calculate the exact cohort sample size, the number of animals used per arm in each experiment was estimated based on variability of pilot and previous experiments. Similarly aged animals were transplanted with the same number of donor cells. Mice were randomly allocated to control or experimental conditions and no animal was excluded from the analysis. For collection of in vivo data, investigators were not blinded to group allocation as the experimental design made this not feasible. For data analysis, if feasible, investigators were blinded to the experimental conditions when assessing the outcome. Tumour-free survival was monitored by Kaplan-Meier analysis, and *p*-values were calculated using log-rank (Mantel–Cox) test. The investigators were not blinded during data collection or analyses.

## Reporting summary

Further information on research design is available in the Nature Portfolio Reporting Summary linked to this article.

## Data availability

The publicly available dataset used in Fig.1b and Supplementary Fig. 1a–c is available in the EMBL-EBI database under accession code E-MTAB-2581. The remaining data are available within the Article, Supplementary Information or Source Data file. Source data are provided with this paper.

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

## Acknowledgements

We would like to thank all members of the Helgason and Vazquez laboratory for input and advice. We thank all patients and healthy donors who donated samples and the National Health Service (NHS) Greater Glasgow and Clyde Biorepository and A. Hair for sample processing. We thank the Core Services and Advanced Technologies at the Cancer Research UK Scotland Institute (C596/A17196; A31287). We thank S. Tardito for providing the components of Plasmax culture medium. We thank J. Tait-Mulder for providing the HCT116 CTR and SHMT2 KO cell lines. We thank C. J. Eaves for providing NRGW[41] mice and the Cancer Research UK Scotland Institute mouse facility staff for housing of mice and help with xenograft experiments. We thank LifeArc for providing the ULK1 inhibitor, MRT403. MRT403 will be made available to academic researchers by LifeArc upon reasonable request and after completion of a material transfer agreement. We thank J. Jones for reading and editing the manuscript. Schematic graphs in Figs. 1a, g. i, 2a, 3d, 4i, 5a, b, d and 6a–d and Supplementary Fig. 7d were created using BioRender (https://biorender.com). This work was supported by Cancer Research UK Glasgow Centre (A25142), Cancer Research UK (C57352/A29754), The Kay Kendall Leukaemia Fund (KKL1069), Blood Cancer UK (formerly Bloodwise; Ref 18006), The Howat Foundation, Tenovus Scotland and Friends of Paul O'Gorman Leukaemia Research Centre (all to G.V.H.); NHSGGC Endowment Fellowship (GN21ON384) to M.M.Z. and Cancer Research UK to D.S2. (A17196 and A31287) and A.V. (A17196 and A21140). M.M.Z. was supported by a Cancer Research UK PhD scholarship (A25142).

## Author contributions

M.M.Z., A.V and G.V.H. conceived and designed the study. M.M.Z. performed all experiments and analysed data. K.M.R. assisted with stable isotope tracing, primary CD34+ CFC experiments and contributed to data interpretation. D.S1. performed microarray dataset analysis and assisted with data interpretation. E.S. assisted with stable isotope tracing experiments and metabolomic data analysis. A.D. and A.I. provided material support and contributed to data interpretation. K.D. assisted with in vivo experiments. M.C. provided material support. D.S2. performed the mass spectrometry. A.V. performed the GC-MS data analysis for the formate levels. M.M.Z. and G.V.H. wrote the manuscript and all other authors reviewed it.

## Competing interests

M.C. has received research funding from Cyclacel and Incyte, is/has been an advisory board member for Novartis, Incyte, Jazz Pharmaceuticals, Pfizer and Servier, and has received honoraria from Astellas, Novartis, Incyte, Pfizer and Jazz Pharmaceuticals. All other authors declare that they have no competing interests.
