## [Peer Review File · Nature Communications]

REVIEWER COMMENTS

Reviewer #1 (Remarks to the Author):

Study elucidates dependency of CML LSCs on folate-mediated 1C metabolism. Inhibition of mitochondrial 1C metabolism by SHMT1/2 inhibitor through de novo purine synthesis showed cytostatic effect in CML. Moreover, modulation of mTORC1 by purine sensing led to erythroid differentiation. Overall, this study is difficult to follow. Results from individual parts are separated and do not draw comprehensive conclusion which would propose new therapy approach for resistant patients. I would recommend to restructure the paper in a way that it follows the main take home message and to put additional findings into supplementary.

Major comments:

1. Identification of leukemic stem cells by CD34 and CD38 is not sufficient since at diagnosis patient in chronic phase of CML still possess normal HSCs as well. Did the authors apply for identification and/or sorting more specific markers, for example CD25, CD26, IL1RAB? The other option is to clarify that CD34+CD38- cells are leukemic by detection of BCR-ABL fusion gene. If not, they should rephrase the type of cells they characterized and studied.
 2. Explain the parallel between K562 which are derived from the patient in blastic crisis and primary CML CD34+CD38- cells.
 3. Experiments on one model cell line, especially K562 are not sufficient in my opinion. Authors used another CML cell line, KCL22 only in in vivo experiment. I would recommend repeating key experiments also in this cell line.
 4. K562 cell line has erythroleukemia features. It is a good model for differentiation studies into erythroid lineage. Authors used those properties to show that SHIN1, inhibitor of mitochondrial 1C metabolism triggered K562 cells to differentiate. They also showed that after folate inhibition, CML CD34+ cells derived from patients increased erythroid markers when cultured with specific supplements. Authors should discuss the rationale of differentiation therapy in chronic phase of the disease and to point out the benefit for CML patients.
 5. Authors emphasize that CD34+CD38- cells are dependent on mitochondrial folate 1C metabolism which is based (if I am not wrong) mainly on gene expression data (Extended Fig.1c). Knockout SHMT1 or some other similar approach is needed to exclude the role of cytosolic folate pathway. SHIN1 inhibits both, SHMT1 and 2 so there is a possibility that cytoplasmic 1C metabolism will also play a role.
 6. Authors claim that impairment of folate metabolism could prime LSCs from quiescent state. This is actually very interesting statement which could be tested in this mouse model. They should characterize changes in differentiation of CD34 cells after IM, SHIN1 and IM+SHIN1 just prior the transplantation into the mice. It could give them better understanding how SHIN1 affects the cells and sensitize them to TKI.
- Results in Figure 6f are in my opinion misinterpreted.

Reviewer #2 (Remarks to the Author):

The present manuscript shows that SHMT inhibition can promote differentiation of CML cells, in particular along the erythroid lineage. It also provides a useful mechanistic dissection implicating mTOR sensing of purines, rather than AMPK, as the main sensing mechanism. Overall, the paper is a good read and contribution to the field.

Major concerns:

1. The final figure involves in vitro treatment of cells prior to their implantation in vivo. This is not what a reader expects based on the abstract description: "Of clinical relevance, we identify that combination of 1C metabolism inhibition with imatinib, a frontline treatment for CML patients, decreases the number of therapy-resistant CML LSCs in a patient-derived xenograft model." The authors are encouraged to use SHIN2, which has OK IV/IP PK, and test its in vivo effects (together with imatinib). Even if they happen to not be impressive, this is important information for the field. If they are impressive, it would greatly enhance the paper. At a bare minimum, if the authors do not

carry out in vivo experiments with SHMT inhibition, the abstract must be clear on this.

2. It seems that AICAR promotes differentiation even in the absence of AMPK? Am I understanding 3g correctly? If so, any ideas/explanation?

Straightforward fixes:

A. The authors claim several places that cytosolic 1C metabolism does not compensate for the loss of SHMT2 in CML. While compensation in vivo for tumorigenesis does seem to be less complete than seen with solid tumors, some cancers seem to form without SHMT2. Have the authors analyze those cells to see whether they reexpress SHMT2? If not, have analyzed whether they use SHMT1 to compensate? Absent careful experimentation, it would be better to avoid over claiming about the lack of SHMT1 role or differences from prior work.

B. Line 137: Clarify ACC1?

C: In general, the discussion lacks the careful precision of the results section. Even though it is discussion, avoid overclaiming. For example, SHIN1 would likely have effects on normal CD34+ cells, even if the duration/dose/assays here did not reveal what these are. Please check whole Discussion to avoid excessive or overly broad claims.

Reviewer #3 (Remarks to the Author):

In this study, Zarou et al investigate SHMT inhibition in CML and show that it drives erythroid differentiation through mTOR mediated purine sensing. Authors reveal that folate metabolism is unregulated in CML LSCs and SHMT inhibition drives cytostatic effects by a block in S phase, as previously shown in other heme malignancies. Authors then show that changes in purine nucleotides drives AMPK activation and reduction in mTOR, leading to erythroid differentiation. Interestingly, the differentiation phenotype is AMPK-independent. Finally, authors show that SHMT inhibition in combination with TKI Imatinib ex vivo leads to a reduction in LSC numbers in a xenograft model. I have several comments for authors to address (not listed in order of importance):

1. Most of the studies are done in a single cell line, K562. I think it would be warranted that authors recapitulate the main phenotypes observed in 1-2 additional cell lines.

2. Related to my previous point, the only in vivo experiment with cell lines (Fig 1h) actually uses KCL22 cells. It is not clear to me why they used these cells instead of K562, and why they didn't use KCL22 cells in any other experiment throughout the paper. Also, the SHMT2 KO levels in KCL22 cells should be shown somewhere (as of now, no WB for these cells is shown).

3. Authors fail to cite very relevant publications on the antileukemic effects of SHMT inhibition, including PMIDs: 32382081 or 34624079, and 34341479 is cited for a marginal reason only. While it is true that SHIN1 is not a good compound for in vivo studies described in the original report, PMID 34341479 used it in vivo and showed antileukemic effects in T-ALL. In addition, PMID 32382081 described SHIN2, a newer version of the compound that shows good PK/PD for in vivo studies. These should all have been described and an in vivo experiment treating mice (xenografted with CML cell lines and/or CML PDXs) with either SHIN1 or SHIN2 should have been done.

4. PMID 34624079 already described that SHMT inhibition leads to mTOR inhibition and increased autophagy In Burkitt Lymphoma, but this is not cited.

5. Authors state they have found an important dependency on the folate pathway for CML LSCs, however, this is never properly assessed. Authors show differences in CML LSCs versus normal HSCs, but what about CML LSCs vs CML non-LSCs? This is important to investigate, as it is unclear right now whether CML cells are generally sensitive to one-carbon inhibition (similar to other leukemias), or if CML LSCs are specifically sensitive or more sensitive than the bulk of CML cells.

6. In Fig 1d, authors should also show proliferation curves of UT + F, H or H/T, so that the effect of these metabolites on growth is properly controlled for. Right now it is only shown what they do in the

context of SHIN1 treatment.

7. Fig 1g, cell cycle bars should add up to 100% in all cases, even if with error bars. Please revise.

8. Data on Fig 6f is not convincing. Authors state that combo SHIN1+TKI reduced colony formation in LTC-IC, however, authors present 2 patient samples and 1 of them show actually increased colonies, not fewer. This is not really convincing with the data presented so far.

9. Authors find SHMT inhibition leads to reduced R5P in CML cells, but it was previously shown to lead to either increased (in T-ALL cells) or unchanged (in colon cancer cells) R5P levels (PMID: 32382081). Authors should discuss these discrepancies.

Summary: We would like to thank all the Reviewers for their constructive comments and believe that by addressing them, our manuscript has substantially improved.

Additional cell line models: To expand our main findings to other cell lines than K562 we have repeated key experiment using KCL22 and JURL-MK1 CML cell lines. Briefly, we show that SHIN1 treatment and SHMT2 knockout (KO) in KCL22 and JURL-MK1 cells phenocopies the effect seen in K562 cells, with decreased proliferation in SHMT2 KO cells, which is rescued with formate or hypoxanthine supplementation (**new Extended Data Fig. 2e-j**). Interestingly, using deuterium labelled 2,3,3-²H₃-serine and ¹³C₃¹⁵N₁-serine we uncover that while all three CML cell lines (K562, KCL22, JURL-MK1) switch to cytosolic thymidine (dTTP) production upon loss of mitochondrial SHMT2, they fail to compensate and generate purines through the cytosolic arm of the pathway, which contrasts with HCT116 colorectal carcinoma cell line and previously published paper by the Rabinowitz lab; PMID: 27211901 (**new Figure 1h-j and Extended Data Fig. 2l-o**).

Furthermore, we reveal that similar to what we observed in K562 cells, SHIN1 treatment results in activation of AMPK and suppression of mTORC1 in KCL22 and JURL-MK1 cells (**new Extended Data Fig. 4b,c**), and promotes erythroid differentiation in JURL-MK1 cells, as seen with K562 cells, which is reversed following formate supplementation (**new Extended Data Fig. 4f**), reconstitution of purine levels (**new Extended Data Fig. 5c,f**), and constitutive activation of mTORC1 through TSC2 KO (**new Extended Data Fig. 5j**). We also show that SHIN1-mediated differentiation of AML cells (THP1, MOLM13) is reversed following reconstitution of purine levels (**new Extended data Fig. 5d,g**). Moreover, while we uncovered that TSC2 KO (leads to hyperactivation of mTORC1) diminishes the effect of SHIN1 on mTORC1 activity in CML cells (K562, KCL22, JURL-MK1), this effect is not seen in AML cells where SHIN1 treatment is still able to inhibit mTORC1 activity (THP1, MOLM13), suggesting that the mechanism of SHIN1-induced differentiation of AML cells may not be exactly that same as in CML cells (**new Extended Data Fig. 5k-n**). In line with this, while rapamycin-mediated mTORC1 inhibition led to differentiation of CML cells, this was not observed in AML cells (**new Extended Data Fig. 5i-n**).

Additional *in vivo* experiment: To address the reviewers' comments regarding administration of pharmacological 1C metabolism inhibitor *in vivo*, we explored the option of treating mice with SHIN2 as Reviewers 2 (Comment 1) and 3 (Comment 3) suggested. However, following investigation of the total cost for sufficient SHIN2 required to conduct this experiment (**1g = £24,400** from commercial sources) we did not think this was a justifiable option (and this was agreed by the Editor, Dr. Morales). Therefore, after consultation with Dr. Morales we decided to perform a cell line xenograft experiment with SHIN1 (**1g = £7,500**), taking the limited bioavailability and potential toxicity into account when performing this experiment.

Overall, although we experienced difficulties with keeping SHIN1 in solution (see more details in response to Reviewer 2, Comment 1), we observed that the combination of SHIN1 and imatinib significantly reduced tumour burden and prolonged survival of treated mice compared to untreated or imatinib only treated cohorts (**new Extended Data Fig. 8g-i**).

Below we provide point-by-point responses to all three Reviewers.

Reviewer #1 (Remarks to the Author):

Study elucidates dependency of CML LSCs on folate-mediated 1C metabolism. Inhibition of mitochondrial 1C metabolism by SHMT1/2 inhibitor through de novo purine synthesis showed cytostatic effect in CML. Moreover, modulation of mTORC1 by purine sensing led to erythroid differentiation. Overall, this study is difficult to follow. Results from individual parts are separated and do not draw comprehensive conclusion which would propose new therapy approach for resistant patients. I would recommend to restructure the paper in a way that it follows the main take home message and to put additional findings into supplementary.

Response: We thank this reviewer for their constructive feedback. We have now added significant amount of new data and tried hard to maintain a good structure of the manuscript.

Major comments:

1. Identification of leukemic stem cells by CD34 and CD38 is not sufficient since at diagnosis patient in chronic phase of CML still possess normal HSCs as well. Did the authors apply for identification and/or sorting more specific markers, for example CD25, CD26, IL1RAB? The other option is to clarify that CD34+CD38- cells are leukemic by detection of BCR-ABL fusion gene. If not, they should rephrase the type of cells they characterized and studied.

Response: To identify transplantable CML sample requires testing of number of samples as not all primary CML samples will engraft in immunodeficient irradiated mice. For each sample that engrafts we perform FISH to ensure the sample engrafts with Philadelphia positive cells (but not normal cells), which is the case for most samples that show engraftment of 1-5% (“contamination” of normal cells leads to higher engraftment). For the PDX shown in **Figure 6f-i, we selected a patient sample that engrafts with >97% Philadelphia positive cells (Rebuttal Fig. 1).**

Rebuttal Fig. 1: Representative FISH image on primary CML sample used in Figure 6f-i.

2.Explain the parallel between K562 which are derived from the patient in blastic crisis and primary CML CD34+CD38- cells.

Response: The immortalised K562 cell line, as the Reviewer mentions, is derived from a CML patient in blast phase (PMID: 28157673), and therefore lacks many of the characteristics present in primary CML CD34⁺38⁻ cells, such as stem cell properties (i.e., self-renewal) and multilineage differentiation potential. However, K562 cells are susceptible to genetic manipulation, target validation, and are commonly used in CML research, contributing to the understanding of fundamental and biological processes, including erythroid differentiation, as well as translational research. Therefore, we have applied it, in addition to other cell line models, to further our understanding of the role 1C metabolism plays in CML cells.

3.Experiments on one model cell line, especially K562 are not sufficient in my opinion. Authors used another CML cell line, KCL22 only in in vivo experiment. I would recommend repeating key experiments also in this cell line.

Response: We thank the reviewer for this constructive comment. As mentioned in our summary above, we have now repeated key experiments using KCL22 and JURL-MK1 CML cells. We show that SHIN1 treatment and SHMT2 KO halt proliferation in all three CML cell lines (**Figure 1d,e and Extended Data Fig. 2a-k**). We show that formate or hypoxanthine supplementation restores growth, while thymidine addition is not sufficient to do so. Furthermore, using deuterium labelled 2,3,3-²H₃-serine and ¹³C₃¹⁵N₁-serine, we establish that while CML cells switch to cytosolic thymidine (dTTP) production upon loss of mitochondrial serine catabolism, mitochondrial 1C metabolism is indispensable for purine biosynthesis (**Figure 1h-j and Extended Data Fig. 2l-o**). This phenotype was consistent in all three CML cell lines, implying that mitochondrial 1C metabolism sustains proliferation through purine synthesis in CML cells. Moreover, we demonstrate that SHIN1 treatment also results in activation of AMPK and suppression of mTORC1 in KCL22 and JURL-MK1 cells (**Extended Data Fig. 4b,c**). As JURL-MK1 cells can upregulate expression of GlyA upon treatment with differentiation-inducing agents (PMID: 15770664), we investigated whether SHIN1 treatment would induce levels of the erythroid markers CD71 and GlyA. As we demonstrated in the K562 cells, SHIN1 treatment increases expression of erythroid differentiation markers in JURL-MK1 cells (**Extended Data Fig. 4f**). Moreover, we observed that this phenotype is reversed following formate supplementation (**Extended Data Fig. 4f**), reconstitution of purine levels (**Extended Data Fig. 5c,f**), and reactivation of mTORC1 through TSC2 KO (**Extended Data Fig. 5j**). Of note, loss of TSC2 diminished the effect of SHIN1 on mTORC1 signalling in both JURL-MK1 and KCL22 (**Extended Data Fig. 5i**).

As presented in the first version of the manuscript, THP1 and MOLM13 cells upregulate the monocytic marker CD11b following SHIN1 treatment (**Extended Data Fig. 4g**). We included these cells in our differentiation experiments and found that reconstitution of purine levels reversed myeloid differentiation (**Extended Data Fig. 5d,g**). However, surprisingly SHIN1

suppressed mTORC1 activity and increased cell surface expression of CD11b in THP1 and MOLM13 TSC2 KO cells (**Extended data Fig. 5k-n**). Moreover, while exposure to rapamycin inhibited mTORC1 signalling, it did not alter levels of CD11b in AML control nor TSC2 KO cells (**Extended data Fig. 5k-n**), suggesting that a separate mechanism is involved in role of TSC2 in regulating mTORC1 activity in SHIN1-treated AML cells. Therefore, delineating the interplay between purine sensing, mTORC1 activity and differentiation in AML cells requires further investigation.

4.K562 cell line has erythroleukemia features. It is a good model for differentiation studies into erythroid lineage. Authors used those properties to show that SHIN1, inhibitor of mitochondrial 1C metabolism triggered K562 cells to differentiate. They also showed that after folate inhibition, CML CD34+ cells derived from patients increased erythroid markers when cultured with specific supplements. Authors should discuss the rationale of differentiation therapy in chronic phase of the disease and to point out the benefit for CML patients.

Response: As suggested, and in relation to new Extended Data Fig. 8a,b (see also response to Comment 6), we have now added relevant text in the Discussion section (line 406-414).

“In relation to CML, differentiation therapy, which would force LSCs to terminally differentiate and have a finite lifespan, has gained interest and is considered a promising therapeutical approach⁵⁸⁻⁶³. However, these approaches have not shown a strong therapeutic benefit for CML patients in clinical trials, highlighting the need for the development of more efficient differentiation inducers⁵⁶. We suggest that inhibition of folate metabolism through impairment of purine biosynthesis can prime LSCs out of quiescence and thereby sensitise them to TKI therapy, which preferentially targets more differentiated cells. Although more work is required to confirm this concept, we found that inhibition of 1C metabolism alone or in combination with TKI therapy increased the fraction of CML CD34⁺ cells found in the S-G2-M phase.”

5.Authors emphasize that CD34+CD38- cells are dependent on mitochondrial folate 1C metabolism which is based (if I am not wrong) mainly on gene expression data (Extended Fig.1c). Knockout SHMT1 or some other similar approach is needed to exclude the role of cytosolic folate pathway. SHIN1 inhibits both, SHMT1 and 2 so there is a possibility that cytoplasmic 1C metabolism will also play a role.

Response: We highlight that folate metabolism is upregulated in CML LSCs, and this upregulation may be driven by overexpression of mitochondrial 1C metabolism genes. While it is tempting to conclude so based on our cell line work, we do not claim that LSCs are dependent on mitochondrial 1C metabolism, which would require further genetic approaches in this rare cell population. Indeed, it would be of interest to generate SHMT2 KO CD34⁺CD38- cells and investigate the reversibility of the cytosolic 1C metabolism arm. However, genetic manipulation of these cells is very challenging, and rarely results in complete gene knockdown. However, motivated by this comment, we decided to further

explore the reversibility of the cytosolic 1C pathway in CML SHMT2 KO cells. In the first version of the manuscript, we uncovered that mitochondrial 1C metabolism sustains proliferation of CML cells both *in vitro* and *in vivo*, as loss of SHMT2 phenocopied the effect of SHIN1 treatment. In this revised version, we have used deuterium labelled 2,3,3-²H₃-serine and ¹³C₃¹⁵N₁-serine to investigate whether CML cells can switch to cytosolic 1C unit production upon genetic ablation of SHMT2 (Figure 1h-j and Extended Data Fig. 2I-o). We established for the first time that while CML cells can switch to cytosolic dTTP synthesis, mitochondrial 1C metabolism is indispensable for purine nucleotide synthesis. Conversely, loss of SHMT2 did not change the ability of HCT116 colorectal carcinoma cells to synthesise ATP, suggesting that the dependency on mitochondrial 1C metabolism might be a unique feature of CML or myeloid leukaemias. In line with this, two independent studies have described that inhibition of both mitochondrial and cytosolic 1C metabolism branches is necessary to halt proliferation of T-ALL cells (PMIDs: 32382081; 34341479).

6. Authors claim that impairment of folate metabolism could prime LSCs from quiescent state. This is actually very interesting statement which could be tested in this mouse model. They should characterize changes in differentiation of CD34 cells after IM, SHIN1 and IM+SHIN1 just prior the transplantation into the mice. It could give them better understanding how SHIN1 affects the cells and sensitize them to TKI.

Response: To address this comment we treated CML CD34⁺ cells (n=4 individual patient samples) with imatinib, SHIN1 or a combination of both for 48h and stained them with the proliferation marker Ki67, and the DNA stain DAPI (Extended Data Fig. 8a,b). We hypothesised to observe a reduction in the fraction of Ki67^{low}DAPI^{low} cells that represent the quiescent cells (G₀). Indeed, SHIN1 monotherapy or in combination with imatinib reduced the fraction of cells in the G₀ phase, although this reduction did not reach statistical significance. Reassuringly, we did see a significant increase in cells in the S-G₂-M phase. Although, these data do not prove that inhibition of 1C metabolism leads to increased differentiation of the most primitive LSCs, they do suggest that inhibition of 1C metabolism primes primitive CML cells out of quiescence.

Results in Figure 6f are in my opinion misinterpreted.

Response: In response to this Comment, Comment 8 from Reviewer 3, and in consideration of new data, we have removed Figure 6f from the manuscript to increase clarity.

Reviewer #2 (Remarks to the Author):

The present manuscript shows that SHMT inhibition can promote differentiation of CML cells, in particular along the erythroid lineage. It also provides a useful mechanistic dissection implicating mTOR sensing of purines, rather than AMPK, as the main sensing mechanism. Overall, the paper is a good read and contribution to the field.

Response: We thank this Reviewer for positive comments about our manuscript.

Major concerns:

1. The final figure involves *in vitro* treatment of cells prior to their implantation *in vivo*. This is not what a reader expects based on the abstract description: "Of clinical relevance, we identify that combination of 1C metabolism inhibition with imatinib, a frontline treatment for CML patients, decreases the number of therapy-resistant CML LSCs in a patient-derived xenograft model." The authors are encouraged to use SHIN2, which has OK IV/IP PK, and test its *in vivo* effects (together with imatinib). Even if they happen to not be impressive, this is important information for the field. If they are impressive, it would greatly enhance the paper. At a bare minimum, if the authors do not carry out *in vivo* experiments with SHMT inhibition, the abstract must be clear on this.

Response: As briefly summarised above, to address this Comment and related Comment 3 from Reviewer 3, we initially enquired about the price of SHIN1 that was successfully used *in vivo* (PMID: 34341479) and SHIN2. With our calculations we would require 1g of compound for 21 days of treatment (100 mg/kg per mouse). As the price that we received for SHIN2 was £24,400, we decided (following a consultation with the Editor) to go ahead with less expensive SHIN1 (£7,500). Given the published concerns around bioavailability of SHIN1, we contacted the first author of PMID: 34341479, and received advice about the suitability of SHIN1 *for in vivo* studies. Although the author restated issues with solubility, we agreed to carry out a KCL22 xenograft experiment as suggested by Reviewers 2 and 3. This revealed that combination of SHIN1 with imatinib significantly reduced the tumour burden compared to vehicle or imatinib treated mice, and prolonged survival in relation to mice treated with imatinib monotherapy (Extended data Fig. 8g-i). However, despite these encouraging positive results, we want to highlight that SHIN1 was very difficult to keep in solution (drug was prepared fresh daily followed by 1h sonication) and drug aggregates were found in the abdominal area of treated mice (Rebuttal Fig. 2).

Rebuttal Fig. 2: Postmortem examination of a mouse treated with SHIN1 for 21 days (daily dose of 100 mg/kg per mouse).

2. It seems that AICAR promotes differentiation even in the absence of AMPK? Am I understanding 3g correctly? If so, any ideas/explanation?

Response: Yes, we believe that AICAR induces differentiation through pyrimidine depletion, and this is not dependent on AMPK activation. Indeed, we show that AICAR depletes pyrimidine nucleotides (Fig. 4i) and pyrimidine depletion (using 6-azauridine) leads to differentiation of K562 cells (Fig. 4k). A relevant paper (PMID: 31431503) has also described that AICAR induces differentiation of AML cells through pyrimidine depletion and independently from AMPK activity.

Straightforward fixes:

A. The authors claim several places that cytosolic 1C metabolism does not compensate for the loss of SHMT2 in CML. While compensation in vivo for tumorigenesis does seem to be less complete than seen with solid tumors, some cancers seem to form without SHMT2. Have the authors analyze those cells to see whether they reexpress SHMT2? If not, have analyzed whether they use SHMT1 to compensate? Absent careful experimentation, it would be better to avoid over claiming about the lack of SHMT1 role or differences from prior work.

Response: We thank the reviewer for this constructive comment. We have now analysed the tumour cells obtained from the single mouse that developed a large tumour, and indeed we do see re-expression of SHMT2, suggesting failure of successful CRISPR-Cas9-mediated gene editing in that clone (Rebuttal Fig. 3, left). Here we made certain that these cells represented human cells, as we confirmed expression of human CD45 by flow cytometry (Rebuttal Fig. 3, right).

Rebuttal Figure 3: SHMT2 re-expression in tumour obtained from mouse injected with SHMT2 KO cells.

Furthermore, motivated by this comment (and Comment 5 from Reviewer 1), we decided to further investigate the reversibility of cytosolic 1C metabolism following SHMT2 KO (see also response to Reviewer 1, Comment 5). We uncovered that mitochondrial 1C metabolism is indeed indispensable for purine synthesis in 3 separate CML cell lines, while cytosolic 1C metabolism can compensate for dTTP synthesis (**Fig. 1h-j and Extended Data Fig. 2l-o**).

B. Line 137: Clarify ACC1?

Response: The text now includes the full name (Line 154).

C: In general, the discussion lacks the careful precision of the results section. Even though it is discussion, avoid overclaiming. For example, SHIN1 would likely have effects on normal CD34⁺ cells, even if the duration/dose/assays here did not reveal what these are. Please check whole Discussion to avoid excessive or overly broad claims.

Response: We have examined the discussion section and made sure not to overstate/overclaim. We feel that with the addition of the new data, we have further evidence to support our claims that CML cells may have a unique dependency on mitochondrial 1C metabolism. We have also removed the sentence about normal CD34⁺ cells and replaced with (Line 434):

“Our findings raise the possibility that inhibiting mitochondrial 1C metabolism may be more selective and have less adverse side effects.”

Reviewer #3 (Remarks to the Author):

In this study, Zarou et al investigate SHMT inhibition in CML and show that it drives erythroid differentiation through mTOR mediated purine sensing. Authors reveal that folate metabolism is unregulated in CML LSCs and SHMT inhibition drives cytostatic effects by a block in S phase, as previously shown in other heme malignancies. Authors then show that changes in purine nucleotides drives AMPK activation and reduction in mTOR, leading to erythroid differentiation. Interestingly, the differentiation phenotype is AMPK-independent. Finally, authors show that SHMT inhibition in combination with TKI Imatinib *ex vivo* leads to a reduction in LSC numbers in a xenograft model.

I have several comments for authors to address (not listed in order of importance):

Response: We thank this Reviewer for nicely summarising our novel findings.

1. Most of the studies are done in a single cell line, K562. I think it would be warranted that authors recapitulate the main phenotypes observed in 1-2 additional cell lines.

Response: As summarised above, we have now repeated key experiments in JURL-MK1 and KCL22 cells as well as AML THP1 and MOLM13 cells. For more detailed response to this comment, I would like to refer to our Response to Reviewer 1, Comment 3.

2. Related to my previous point, the only *in vivo* experiment with cell lines (Fig 1h) actually uses KCL22 cells. It is not clear to me why they used these cells instead of K562, and why they didn't use KCL22 cells in any other experiment throughout the paper. Also, the SHMT2 KO levels in KCL22 cells should be shown somewhere (as of now, no WB for these cells is shown).

Response: We have now included a WB showing levels of SHMT2 in KCL22 CTR and SHMT2 KO cells (Extended data Fig. 2f). In our hands K562 cells do not consistently form tumours *in vivo*, while KCL22 are a more reliable model for target assessment studies. As mentioned in my response to Comment 1 (and detailed in Response to Reviewer 1, Comment 3) we have now included new *in vitro* data on KCL22 cells that show similar phenotype following inhibition of 1C metabolism to K562 cells.

3. Authors fail to cite very relevant publications on the antileukemic effects of SHMT inhibition, including PMIDs: 32382081 or 34624079, and 34341479 is cited for a marginal reason only. While it is true that SHIN1 is not a good compound for *in vivo* studies described in the original report, PMID 34341479 used it *in vivo* and showed antileukemic effects in T-ALL. In addition, PMID 32382081 described SHIN2, a newer version of the compound that shows good PK/PD for *in vivo* studies. These should all have been described and an *in vivo* experiment treating mice (xenografted with CML cell lines and/or CML PDXs) with either SHIN1 or SHIN2 should have been done.

Response: We thank the reviewer for highlighting these papers and have now cited all three studies (Lines 342-344; 368-370, 378-380). We have also carried out an *in vivo* experiment with the SHIN1 as suggested. For detailed response to this Comment, I would like to refer to our Summary and Response to Reviewer 2, Comment 1.

4. PMID 34624079 already described that SHMT inhibition leads to mTOR inhibition and increased autophagy In Burkitt Lymphoma, but this is not cited.

Response: This has now been cited (Lines 378-380).

5. Authors state they have found an important dependency on the folate pathway for CML LSCs, however, this is never properly assessed. Authors show differences in CML LSCs versus normal HSCs, but what about CML LSCs vs CML non-LSCs? This is important to investigate, as it is unclear right now whether CML cells are generally sensitive to one-carbon inhibition (similar to other leukemias), or if CML LSCs are specifically sensitive or more sensitive than the bulk of CML cells.

Response: We agree with this comment and throughout the paper have been careful not to claim that this is a specific dependency of CML LSCs. While we demonstrate that CML LSCs are sensitive to SHIN1 treatment (Fig. 6f-i), our data also suggest that the bulk of CD34⁺ (stem/progenitor) cells are also sensitive to SHIN1 treatment (i.e., Fig. 6a-d). Indeed, further analysis comparing LSCs vs LPCs (progenitor cells), suggests that these enzymes are equally expressed in LPCs (Rebuttal Fig. 4):

Rebuttal Fig. 4: Transcriptomic analysis of folate metabolism genes in LSCs and LPCs.

6. In Fig 1d, authors should also show proliferation curves of UT + F, H or H/T, so that the effect of these metabolites on growth is properly controlled for. Right now it is only shown what they do in the context of SHIN1 treatment.

Response: We have now included these data (Extended Data Fig. 2a). As the figure suggests, addition of these treatments in cells with competent 1C metabolism does not significantly alter cell proliferation.

7. Fig 1g, cell cycle bars should add up to 100% in all cases, even if with error bars. Please revise.

Response: This has now been revised and bars add up to 100% (Fig. 1g and Extended Data Fig. 2k).

8. Data on Fig 6f is not convincing. Authors state that combo SHIN1+TKI reduced colony formation in LTC-IC, however, authors present 2 patient samples and 1 of them show actually increased colonies, not fewer. This is not really convincing with the data presented so far.

Response: In response to this Comment, Comment 6 from Reviewer 1, and in consideration of new data, we have removed Figure 6f from the manuscript to increase clarity.

9. Authors find SHMT inhibition leads to reduced R5P in CML cells, but it was previously shown to lead to either increased (in T-ALL cells) or unchanged (in colon cancer cells) R5P levels (PMID: 32382081). Authors should discuss these discrepancies.

Response: We thank the Reviewer for this Comment. We believe that reduced R5P is related to decreased glucose consumption following 1C metabolism inhibition (Fig. 2 and Extended Data Fig. 3). We see an increase in intracellular glucose and decrease in intracellular lactate and pyruvate (Fig. 2d and Extended Data Fig. 3d) as well as reduced glycolysis (Fig. 2f,g and Extended Data Fig. 3f,g). These data suggest that there is less glucose consumption following SHMT inhibition. Less glucose usage would lead to less contribution to pentose phosphate pathway. Indeed, we see a decrease in glucose-6-phosphate following SHIN1 treatment or loss of SHMT2, the glycolysis intermediate that enters the pentose phosphate pathway (Rebuttal Fig. 5). Whether there is feedback between 1C unit availability and glucose consumption in CML cells requires further investigation. However, as we mention in the manuscript (Line 146) our findings are in agreement with our previous work where we applied theoretical modelling to indicate that 1C metabolism promotes glycolysis (PMID: 32366892).

Rebuttal Fig. 5: Glucose-6-phosphate levels in K562 cells, measured by LC-MS, following SHMT2 KO and indicated treatments for 24 h.

REVIEWERS' COMMENTS

Reviewer #1 (Remarks to the Author):

Authors addressed all my comments and clarified their results.

Reviewer #2 (Remarks to the Author):

Given that the experiment in Fig 6f-i is a bit odd, I believe that the new in vivo experiment in Fig S8 (also imperfect in using SHIN1 with bad PK rather than SHIN2) should be put into the main text.

Otherwise, the revision is acceptable.

[Reviewer #3 signs off privately to the Editor]